# Bottleneck Structure in Learned Features: Low-Dimension vs Regularity Tradeoff

**Arthur Jacot**
Courant Institute of Mathematical Sciences
New York University
New York, NY 10012
arthur.jacot@nyu.edu

## Abstract

Previous work [Jac23] has shown that DNNs with large depth $L$ and $L_2$-regularization are biased towards learning low-dimensional representations of the inputs, which can be interpreted as minimizing a notion of rank $R^{(0)}(f)$ of the learned function $f$, conjectured to be the Bottleneck rank. We compute finite depth corrections to this result, revealing a measure $R^{(1)}$ of regularity which bounds the pseudo-determinant of the Jacobian $|Jf(x)|_+$ and is subadditive under composition and addition. This formalizes a balance between learning low-dimensional representations and minimizing complexity/irregularity in the feature maps, allowing the network to learn the 'right' inner dimension. Finally, we prove the conjectured bottleneck structure in the learned features as $L \to \infty$: for large depths, almost all hidden representations are approximately $R^{(0)}(f)$-dimensional, and almost all weight matrices $W_\ell$ have $R^{(0)}(f)$ singular values close to 1 while the others are $O(L^{-\frac{1}{2}})$. Interestingly, the use of large learning rates is required to guarantee an order $O(L)$ NTK which in turns guarantees infinite depth convergence of the representations of almost all layers.

## 1 Introduction

The representation cost $R(f; L) = \min_{\theta: f_\theta = f} \|\theta\|^2$ [DKS21] can be defined for any model and describes the bias in function space resulting from the minimization of the $L_2$-norm of the parameters. While it can be computed explicitly for linear networks [DKS21] or shallow nonlinear ones [Bac17], the deep non-linear case remains ill-understood [JGHG22].

Previous work [Jac23] has shown that the representation cost of DNNs with homogeneous nonlinearity $\sigma$ converges to a notion of rank over nonlinear functions $\lim_{L\to\infty} \frac{R(f;L)}{L} \to R^{(0)}(f)$. Over a large set of functions $f$, the limiting representation cost $R^{(0)}(f)$ was proven the so-called Bottleneck (BN) rank $\text{Rank}_{BN}(f)$ which is the smallest integer $k$ such that $f : \mathbb{R}^{d_{in}} \to \mathbb{R}^{d_{out}}$ can be factored $f = \mathbb{R}^{d_{in}} \xrightarrow{g} \mathbb{R}^k \xrightarrow{h} \mathbb{R}^{d_{out}}$ with inner dimension $k$ (it is conjectured to match it everywhere). This suggests that large depth $L_2$-regularized DNNs are adapted for learning functions of the form $f^* = g \circ h$ with small inner dimension.

This can also be interpreted as DNNs learning symmetries, since a function $f : \Omega \to \mathbb{R}^{d_{out}}$ with symmetry group $G$ (i.e. $f(g \cdot x) = f(x)$) can be defined as mapping the inputs $\Omega$ to an embedding of the modulo space $\Omega/G$ and then to the outputs $\mathbb{R}^{d_{out}}$. Thus a function with a lot of symmetries will have a small BN-rank, since $\text{Rank}_{BN}(f) \leq \dim(\Omega/G)$ where $\dim(\Omega/G)$ is the smallest dimension $\Omega/G$ can be embedded into.

37th Conference on Neural Information Processing Systems (NeurIPS 2023).

A problem is that this notion of rank does not control the regularity of $f$, but results of [Jac23] suggest that a measure of regularity might be recovered by studying finite depths corrections to the $R^{(0)}$ approximation. This formalizes the balance between minimizing the dimension of the learned features and their complexity.

Another problem is that minimizing the rank $R^{(0)}$ does not uniquely describe the learned function, as there are many fitting functions with the same rank. Corrections allow us to identify the learned function amongst those.

Finally, the theoretical results and numerical experiments of [Jac23] strongly suggest a bottleneck structure in the learned features for large depths, where the (possibly) high dimensional input data is mapped after a few layers to a low-dimensional hidden representation, and keeps the approximately same dimensionality until mapping back to the high dimensional outputs in the last few layers. We prove the existence of such a structure, but with potentially multiple bottlenecks.

## 1.1 Contributions

We analyze the Taylor approximation of the representation cost around infinite depth $L = \infty$:

$$R(f; \Omega, L) = LR^{(0)}(f; \Omega) + R^{(1)}(f; \Omega) + \frac{1}{L}R^{(2)}(f; \Omega) + O(L^{-2}).$$

The first correction $R^{(1)}$ measures some notion of regularity of the function $f$ that behaves sub-additively under composition $R^{(1)}(f \circ g) \leq R^{(1)}(f) + R^{(1)}(g)$ and under addition $R^{(1)}(f + g) \leq R^{(1)}(f) + R^{(1)}(g)$ (under some constraints), and controls the Jacobian of $f$: $\forall x, 2 \log |Jf(x)|_+ \leq R^{(1)}(f)$, where $|\cdot|_+$ is the *pseudo-determinant*, the product of the non-zero singular values.

This formalizes the balance between the bias towards minimizing the inner dimension described by $R^{(0)}$ and a regularity measure $R^{(1)}$. As the depth $L$ grows, the low-rank bias $R^{(0)}$ dominates, but even in the infinite depth limit the regularity $R^{(1)}$ remains relevant since there are typically multiple fitting functions with matching $R^{(0)}$ which can be differentiated by their $R^{(1)}$ value.

For linear networks, the second correction $R^{(2)}$ guarantees infinite depth convergence of the representations of the network. We recover several properties of $R^{(2)}$ in the nonlinear case, but we also give a counterexample that shows that norm minimization does not guarantee convergence of the representation, forcing us to look at other sources of bias.

To solve this issue, we show that a $\Theta(L^{-1})$ learning rate forces the NTK to be $O(L)$ which in turn guarantees the convergence as $L \to \infty$ of the representations at almost every layer of the network.

Finally we prove the Bottleneck structure that was only observed empricially in [Jac23]: we show that the weight matrices $W_\ell$ are approximately rank $R^{(0)}(f)$, more precisely $W_\ell$ has $R^{(0)}(f)$ singular values that are $O(L^{-\frac{1}{2}})$ close to 1 and all the other are $O(L^{-\frac{1}{2}})$. Together with the $O(L)$ NTK assumption, this implies that the pre-activations $\alpha_\ell(X)$ of a general dataset at almost all layer is approximately $R^{(0)}(f)$-dimensional, more precisely that the $k + 1$-th singular value of $\alpha_\ell(X)$ is $O(L^{-\frac{1}{2}})$.

## 1.2 Related Works

The representation cost has mostly been studied in settings where an explicit formula can be obtained, such as in linear networks [DKS21], or shallow nonlinear networks [Bac17], or for deep networks with very specific structure [OW22, LJ22]. A low rank phenomenon in large depth $L_2$-regularized DNNs has been observed in [TVS22].

Regarding deep fully-connected networks, two reformulations of the representation cost optimization have been given in [JGHG22], which also shows that the representation becomes independent of the width as long as the width is large enough.

The Bottleneck structure that we describe in this paper is similar to the Information Bottleneck theory [TZ15]. It is not unlikely that the bias towards dimension reduction in the middle layers of the network could explain the loss of information that was observed in the first layers of the network in [TZ15].

## 2 Setup

In this paper, we study fully connected DNNs with $L + 1$ layers numbered from $0$ (input layer) to $L$ (output layer). The layer $\ell \in \{0, \dots, L\}$ has $n_\ell$ neurons, with $n_0 = d_{in}$ the input dimension and $n_L = d_{out}$ the output dimension. The pre-activations $\tilde{\alpha}_\ell(x) \in \mathbb{R}^{n_\ell}$ and activations $\alpha_\ell(x) \in \mathbb{R}^{n_\ell}$ are defined by

$$
\begin{aligned}
\alpha_0(x) &= x \\
\tilde{\alpha}_\ell(x) &= W_\ell \alpha_{\ell-1}(x) + b_\ell \\
\alpha_\ell(x) &= \sigma\left(\tilde{\alpha}_\ell(x)\right),
\end{aligned}
$$

for the $n_\ell \times n_{\ell-1}$ connection weight matrix $W_\ell$, the $n_\ell$-dim bias vector $b_\ell$ and the nonlinearity $\sigma : \mathbb{R} \to \mathbb{R}$ applied entry-wise to the vector $\tilde{\alpha}_\ell(x)$. The parameters of the network are the collection of all connection weights matrices and bias vectors $\theta = (W_1, b_1, \dots, W_L, b_L)$. The network function $f_\theta : \mathbb{R}^{d_{in}} \to \mathbb{R}^{d_{out}}$ is the function that maps an input $x$ to the pre-activations of the last layer $\tilde{\alpha}_L(x)$.

We assume that the nonlinearity is of the form $\sigma_a(x) = \begin{cases} x & \text{if } x \geq 0 \\ ax & \text{otherwise} \end{cases}$ for some $\alpha \in (-1, 1)$ (yielding the ReLU for $\alpha = 0$), as any homogeneous nonlinearity $\sigma$ (that is not proportional to the identity function, the constant zero function or the absolute function) matches $\sigma_a$ up to scaling and inverting the inputs.

The functions that can be represented by networks with homogeneous nonlinearities and any finite depth/width are exactly the set of finite piecewise linear functions (FPLF) [ABMM18, HLXZ18].

*Remark* 1. In most of our results, we assume that the width is sufficiently large so that the representation cost matches the infinite-width representation cost. For a dataset of size $N$, a width of $N(N + 1)$ suffices, as shown in [JGHG22] (though a much smaller width is often sufficient).

### 2.1 Representation Cost

The representation cost $R(f; \Omega, L)$ is the minimal squared parameter norm required to represent the function $f$ over the input domain $\Omega$:

$$
R(f; \Omega, L) = \min_{\theta : f_{\theta|\Omega} = f_{|\Omega}} \|\theta\|^2
$$

where the minimum is taken over all weights $\theta$ of a depth $L$ network (with some finite widths $n_1, \dots, n_{L-1}$) such that $f_\theta(x) = f(x)$ for all $x \in \Omega$. If no such weights exist, we define $R(f; \Omega, L) = \infty$.

The representation cost describes the natural bias on the represented function $f_\theta$ induced by adding $L_2$ regularization on the weights $\theta$:

$$
\min_\theta C(f_\theta) + \lambda \|\theta\|^2 = \min_f C(f) + \lambda R(f; \Omega, L)
$$

for any cost $C$ (defined on functions $f : \Omega \mapsto \mathbb{R}^{d_{out}}$) and where the minimization on the right is over all functions $f$ that can be represented by a depth $L$ network with nonlinearity $\sigma$.

For any two functions $f, g$, we denote $f \to g$ the function $h$ such that $g = h \circ f$, assuming it exists, and we write $R(f \to g; \Omega, L) = R(h; f(\Omega), L)$.

*Remark* 2. The representation cost also describes the implicit bias of networks trained with the cross-entropy loss [SHN$^+$18, GLSS18, CB20].

## 3 Representation Cost Decomposition

Since there are no explicit formula for the representation cost of deep nonlinear networks, we propose to approximate it by a Taylor decomposition in $1/L$ around $L = \infty$. This is inspired by the behavior of the representation cost of deep linear networks (which represent a matrix as a product $A_\theta = W_L \cdots W_1$), for which an explicit formula exists [DKS21]:

$$
R(A; L) = \min_{\theta : A = A_\theta} \|\theta\|^2 = L \|A\|_{2/L}^{2/L} = L \sum_{i=1}^{\text{Rank} A} s_i(A)^{2/L},
$$

where $\|\cdot\|_p^p$ is the $L_p$-Schatten norm, the $L_p$ norm on the singular values $s_i(A)$ of $A$.

Approximating $s^{\frac{2}{L}} = 1 + \frac{2}{L}\log s + \frac{2}{L^2}(\log s)^2 + O(L^{-3})$, we obtain

$$R(A; L) = L\mathrm{Rank}A + 2\log|A|_+ + \frac{1}{2L}\left\|\log_+ A^T A\right\|^2 + O(L^{-2}),$$

where $\log_+$ is the pseudo-log, which replaces the non-zero eigenvalues of $A$ by their log.

We know that gradient descent will converge to parameters $\theta$ representing a matrix $A_\theta$ that locally minimize the loss $C(A) + \lambda R(A; L)$. The approximation $R(A; L) \approx L\mathrm{Rank}A$ fails to recover the local minima of this loss, because the rank has zero derivatives almost everywhere. But this problem is alleviated with the second order approximation $R(A; L) \approx L\mathrm{Rank}A + 2\log|A|_+$. The minima can then be interpreted as first minimizing the rank, and then choosing amongst same rank solutions the matrix with the smallest pseudo-determinant. Changing the depth allows one to tune the balance between minimizing the rank and the regularity of the matrix $A$.

## 3.1 First Correction: Regularity Control

As a reminder, the dominating term in the representation cost $R^{(0)}(f)$ is conjectured in [Jac23] to converge to the so-called Bottleneck rank $\mathrm{Rank}_{BN}(f; \Omega)$ which is the smallest integer $k$ such that $f$ can be decomposed as $f = \Omega \xrightarrow{g} \mathbb{R}^k \xrightarrow{h} \mathbb{R}^{d_{out}}$ with inner dimension $k$, and where $g$ and $h$ are FPLF. A number of results supporting this conjecture are proven in [Jac23]: a sandwich bound

$$\mathrm{Rank}_J(f; \Omega) \leq R^{(0)}(f; \Omega) \leq \mathrm{Rank}_{BN}(f; \Omega)$$

for the Jacobian rank $\mathrm{Rank}_J(f; \Omega) = \max_x \mathrm{Rank} Jf(x)_{|T_x\Omega}$, and three natural properties of ranks that $R^{(0)}$ satisfies:

1. $R^{(0)}(f \circ g; \Omega) \leq \min\left\{R^{(0)}(f), R^{(0)}(g)\right\}$,
2. $R^{(0)}(f + g; \Omega) \leq R^{(0)}(f) + R^{(0)}(g)$,
3. $R^{(0)}(x \mapsto Ax; \Omega) = \mathrm{Rank}A$ for any full dimensional and bounded $\Omega$.

These results imply that for any function $f = \phi \circ A \circ \psi$ that is linear up to bijections $\phi, \psi$, the conjecture is true $R^{(0)}(f; \Omega) = \mathrm{Rank}_{BN}(f; \Omega) = \mathrm{Rank}A$.

The proof of the aforementioned sandwich bound in [Jac23] actually prove an upper bound of the form $L\mathrm{Rank}_{BN}(f; \Omega) + O(1)$ thus proving that the $R^{(1)}$ term is upper bounded. The following theorem proves a lower bound on $R^{(1)}$ as well as some of its properties:

**Theorem 3.** *For all inputs $x$ where $\mathrm{Rank}Jf(x) = R^{(0)}(x)$, $R^{(1)}(f) \geq 2\log|Jf(x)|_+$, furthermore:*

1. *If $R^{(0)}(f \circ g) = R^{(0)}(f) = R^{(0)}(g)$, then $R^{(1)}(f \circ g) \leq R^{(1)}(f) + R^{(1)}(g)$.*
2. *If $R^{(0)}(f + g) = R^{(0)}(f) + R^{(0)}(g)$, then $R^{(1)}(f + g) \leq R^{(1)}(f) + R^{(1)}(g)$.*
3. *If $P_{\mathrm{Im}A^T}\Omega$ and $A\Omega$ are $k = \mathrm{Rank}A$ dimensional and completely positive (i.e. they can be embedded isometrically into $\mathbb{R}^m_+$ for some $m$), then $R^{(1)}(x \mapsto Ax; \Omega) = 2\log|A|_+$.*

Notice how these properties clearly point to the first correction $R^{(1)}(f)$ measuring a notion of regularity of $f$ instead of a notion of rank. One can think of $L_2$-regularized deep nets as learning functions $f$ that minimize

$$\min_f C(f(X)) + \lambda L R^{(0)}(f) + \lambda R^{(1)}(f).$$

The depth determines the balance between the rank regularization and regularity regularization. Without the $R^{(1)}$-term, the above minimization would never be unique since there can be multiple functions $f$ with the same training outputs $f(X)$ with the same rank $R^{(0)}(f)$.

Under the assumption that $R^{(0)}$ only takes integer value the above optimization can be rewritten as

$$\min_{k=0,1,\dots} \lambda L k + \min_{f:R^{(0)}(f)=k} C(f(X)) + \lambda R^{(1)}(f).$$

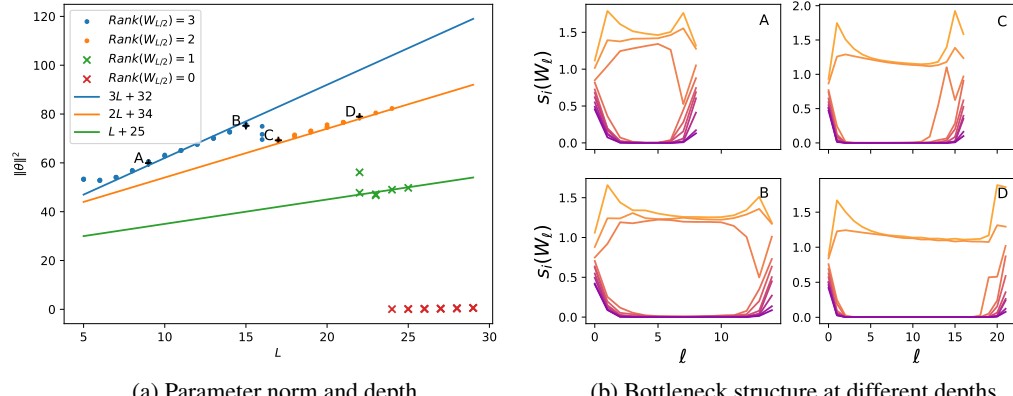

(a) Parameter norm and depth                    (b) Bottleneck structure at different depths.

Figure 1: (a) Plot of the parameter norm at the end of training ($\lambda = 0.001$) over a range of depths, colored acoording to the rank (# of sing. vals above 0.1) of the weight matrices $W_{L/2}$ in the middle of the network, and marked with a dot '.' or cross 'x' depending on whether the final train cost is below or above 0.1. The training data is synthetic and designed to have a optimal rank $k^* = 2$. We see different ranges of depth where the network converges to different rank, with larger depths leading to smaller rank, until training fails and recover the zero parameters for $L > 25$. Within each range the norm $\|\theta\|^2$ is well approximated by a affine function with slope equal to the rank. (b) Plot of the singular values of $W_\ell$ throughout the networks for 4 trials, we see that the bottleneck structure remains essentially the same throughout each range of depth, with only the middle low-rank part growing with the depth.

Every inner minimization for a rank $k$ that is attained inside the set $\{f : R^{(0)}(f) = k\}$ corresponds to a different local minimum. Note how these inner minimization do not depend on the depth, suggesting the existence of sequences of local minima for different depths that all represent approximately the same function $f$, as can be seen in Figure 1. We can classify these minima according to whether they recover the true BN-rank $k^*$ of the task, underestimate or overestimate it.

In linear networks [WJ23], rank underestimating minima cannot fit the data, but it is always possible to fit any data with a BN-rank 1 function (by mapping injectively the datapoints to a line and then mapping them nonlinearly to any other configuration). We therefore need to also differentiate between rank-underestimating minima that fit or do not fit the data. The non-fitting minima can in theory be avoided by taking a small enough ridge (along the lines of [WJ23]), but we do observe them emprically for large depths in Figure 1.

In contrast, we have never observed fitting rank-underestimating minima, though their existence was proven for large enough depths in [Jac23]. A possible explanation for why GD avoids these minima is their $R^{(1)}$ value explodes with the number of datapoints $N$, since these network needs to learn a space filling surface (a surface of dimension $k < k^*$ that visits random outputs $y_i$ that are sampled from a $k^*$-dimensional distribution). More precisely Theorem 2 of [Jac23] implies that the $R^{(1)}$ value of fitting BN-rank 1 minima explodes at a rate of $2(1 - \frac{1}{k^*}) \log N$ as the number of datapoints $N$ grows, which could explain why we rarely observe such minima in practice, but another explanation could be that these minima are very narrow, as explained in Section 4.1.

In our experiments we often encountered rank overestimating minima and we are less sure about how to avoid them, though it seems that increasing the depth helps (see Figure 1), and that SGD might help too by analogy with the linear case [WJ23]. Thankfully overestimating the rank is less problematic for generalization, as supported by the fact that it is possible to approximate BN-rank $k^*$ with a higher rank function with any accuracy, while doing so with a low rank function requires a pathological function.

## 3.2 Second Correction

We now identify a few properties of the second correction $R^{(2)}$:

**Proposition 4.** *If there is a limiting representation as $L \to 0$ in the optimal representation of $f$, then $R^{(2)}(f) \geq 0$. Furthermore:*

1. *If $R^{(0)}(f \circ g) = R^{(0)}(f) = R^{(0)}(g)$ and $R^{(1)}(f \circ g) = R^{(1)}(f) + R^{(1)}(g)$, then*
   $$\sqrt{R^{(2)}(f \circ g)} \leq \sqrt{R^{(2)}(f)} + \sqrt{R^{(2)}(g)}.$$

2. *If $R^{(0)}(f + g) = R^{(0)}(f) + R^{(0)}(g)$ and $R^{(1)}(f + g) = R^{(1)}(f) + R^{(1)}(g)$, then $R^{(2)}(f + g) \leq R^{(2)}(f) + R^{(2)}(g)$.*

3. *If $A^p \Omega$ is $k = \text{Rank} A$-dimensional and completely positive for all $p \in [0, 1]$, where $A^p$ has its non-zero singular taken to the $p$-th power, then $R^{(2)}(x \mapsto Ax; \Omega) = \frac{1}{2} \left\| \log_+ A^T A \right\|^2$.*

While the properties are very similar to those of $R^{(1)}$, the necessary conditions necessary to apply them are more restrictive. There might be case where the first two terms $R^{(0)}$ and $R^{(1)}$ do not uniquely determine a minimum, in which case the second correction $R^{(2)}$ needs to be considered.

In linear networks the second correction $R^{(2)}(A) = \frac{1}{2} \left\| \log_+ A^T A \right\|^2$ plays an important role, as it bounds the operator norm of $A$ (which is not bounded by $R^{(0)}(A) = \text{Rank} A$ nor $R^{(1)}(A) = 2 \log |A|_+$), thus guaranteeing the convergence of the hidden representations in the middle of network. We hoped at first that $R^{(2)}$ would have similar properties in the nonlinear case, but we were not able to prove anything of the sort. Actually in contrast to the linear setting, the representations of the network can diverge as $L \to \infty$, which explains why the $R^{(2)}$ does not give any similar control, which would guarantee convergence.

## 3.3 Representation geodesics

One can think of the sequence of hidden representations $\tilde{\alpha}_1, \ldots, \tilde{\alpha}_L$ as a path from the input representation to the output representation that minimizes the weight norm $\|W_\ell\|^2$ required to map from one representation to the next. As the depth $L$ grows, we expect this sequence to converge to a form of geodesic in representation space. Such an analysis has been done in [Owh20] for ResNet where these limiting geodesics are continuous.

Two issues appear in the fully-connected case. First a representation $\tilde{\alpha}_\ell$ remains optimal after any swapping of its neurons or other symmetries, but this can easily be solved by considering representations $\tilde{\alpha}_\ell$ up to orthogonal transformation, i.e. to focus on the kernels $K_\ell(x, y) = \tilde{\alpha}_\ell(x)^T \tilde{\alpha}_\ell(y)$. Second the limiting geodesics of fully-connected networks are not continuous, and as such they cannot be described by a local metric.

We therefore turn to the representation cost of DNNs to describe the hidden representations of the network, since the $\ell$-th pre-activation function $\tilde{\alpha}^{(\ell)} : \Omega \to \mathbb{R}^{n_\ell}$ in a network which minimizes the parameter norm must satisfy

$$R(f; \Omega, L) = R(\tilde{\alpha}_\ell; \Omega, \ell) + R(\sigma(\tilde{\alpha}_\ell) \to f; \Omega, L - \ell).$$

Thus the limiting representations $\tilde{\alpha}_p = \lim_{L \to \infty} \tilde{\alpha}_{\ell_L}$ (for a sequence of layers $\ell_L$ such that $\lim_{L \to \infty} \ell_L / L = p \in (0, 1)$) must satisfy

$$R^{(0)}(f; \Omega) = R^{(0)}(\tilde{\alpha}_p; \Omega) = R^{(0)}(\sigma(\tilde{\alpha}_p) \to f; \Omega)$$
$$R^{(1)}(f; \Omega) = R^{(1)}(\tilde{\alpha}_p; \Omega) + R^{(1)}(\sigma(\tilde{\alpha}_p) \to f; \Omega)$$

Let us now assume that the limiting geodesic is continuous at $p$ (up to orthogonal transformation, which do not affect the representation cost), meaning that any other sequence of layers $\ell'_L$ converging to the same ratio $p \in (0, 1)$ would converge to the same representation. The taking the limits with two sequences $\lim \frac{\ell_L}{L} = p = \lim \frac{\ell'_L}{L}$ such that $\lim \ell'_L - \ell_L = +\infty$ and and taking the limit of the equality

$$R(f; \Omega, L) = R(\tilde{\alpha}_{\ell_L}; \Omega, \ell_L) + R(\sigma(\tilde{\alpha}_{\ell_L}) \to \tilde{\alpha}_{\ell'_L}; \Omega, \ell'_L - \ell_L) + R(\sigma(\tilde{\alpha}_{\ell'_L}) \to f; \Omega, L - \ell'_L),$$

we obtain that $R^{(0)}(\sigma(\tilde{\alpha}_p) \to \tilde{\alpha}_p; \Omega) = R^{(0)}(f)$ and $R^{(1)}(\sigma(\tilde{\alpha}_p) \to \tilde{\alpha}_p; \Omega) = 0$. This implies that $\sigma(\tilde{\alpha}(x)) = \tilde{\alpha}(x)$ at any point $x$ where $\mathrm{Rank}\,Jf(x) = R^{(0)}(f; \Omega)$, thus $R^{(0)}(id; \tilde{\alpha}_p(\Omega)) = R^{(0)}(f; \Omega)$ and $R^{(1)}(id; \tilde{\alpha}_p(\Omega)) = 0$ if $\mathrm{Rank}\,Jf(x) = R^{(0)}(f; \Omega)$ for all $x \in \Omega$.

### 3.3.1 Identity

When evaluated on the identity, the first two terms $R^{(0)}(id; \Omega)$ and $R^{(1)}(id; \Omega)$ describe properties of the domain $\Omega$.

For any notion of rank, $\mathrm{Rank}(id; \Omega)$ defines a notion of dimensionality of $\Omega$. The Jacobian rank $\mathrm{Rank}_J(id; \Omega) = \max_{x \in \Omega} \dim T_x \Omega$ is the maximum tangent space dimension, while the Bottleneck rank $\mathrm{Rank}_{BN}(id; \Omega)$ is the smallest dimension $\Omega$ can be embedded into. For example, the circle $\Omega = \mathbb{S}^{2-1}$ has $\mathrm{Rank}_J(id; \Omega) = 1$ and $\mathrm{Rank}_{BN}(id; \Omega) = 2$.

On a domain $\Omega$ where the two notions of dimensionality match $\mathrm{Rank}_J(id; \Omega) = \mathrm{Rank}_{BN}(id; \Omega) = k$, the first correction $R^{(1)}(id; \Omega)$ is non-negative since for any $x$ with $\dim T_x \Omega = k$, we have $R^{(1)}(id; \Omega) \geq \log |P_{T_x}|_+ = 0$. The $R^{(1)}(id; \Omega)$ value measures how non-planar the domain $\Omega$ is, being 0 only if $\Omega$ is $k$-planar, i.e. its linear span is $k$-dimensional:

**Proposition 5.** *For a domain with* $\mathrm{Rank}_J(id; \Omega) = \mathrm{Rank}_{BN}(id; \Omega) = k$, *then* $R^{(1)}(id; \Omega) = 0$ *if and only if* $\Omega$ *is* $k$-*planar and completely positive.*

This proposition shows that the $R^{(1)}$ term does not only bound the Jacobian of $f$ as shown in Theorem 3, but also captures properties of the curvature of the domain/function.

Thus at ratios $p$ where the representation geodesics converge continuously, the representations $\tilde{\alpha}_p(\Omega)$ are $k = R^{(0)}(f; \Omega)$-planar, proving the Bottleneck structure that was only observed empirically in [Jac23]. But the assumption of convergence over which we build this argument does not hold in general, actually we give in the appendix an example of a simple function $f$ whose optimal representations diverges in the infinite depth limit. This is in stark contrast to the linear case, where the second correction $R^{(2)}(A) = \frac{1}{2} \left\| \log_+ A^T A \right\|^2$ guarantees convergence, since it bounds the operator norm of $A$. To prove and describe the bottleneck structure in nonlinear DNNs, we therefore need to turn to another strategy.

## 4 Bottleneck Structure in Large Depth Networks

Up to now we have focused on one aspect of the Bottleneck structure observed in [Jac23]: that the representations $\alpha_\ell(X)$ inside the Bottleneck are approximately $k$-planar. But another characteristic of this phenomenon is that the weight matrices $W_\ell$ in the bottleneck have $k$ dominating singular values, all close to 1. This property does not require the convergence of the geodesics and can be proven with finite depth rates:

**Theorem 6.** *Given parameters* $\theta$ *of a depth* $L$ *network, with* $\|\theta\|^2 \leq kL + c_1$ *and a point* $x$ *such that* $\mathrm{Rank}\,Jf_\theta(x) = k$, *then there are* $w_\ell \times k$ *(semi-)orthonormal* $V_\ell$ *such that* $\sum_{\ell=1}^{L} \left\| W_\ell - V_\ell V_{\ell-1}^T \right\|_F^2 \leq c_1 - 2 \log |Jf_\theta(x)|_+$ *thus for any* $p \in (0,1)$ *there are at least* $(1-p)L$ *layers* $\ell$ *with*

$$\left\| W_\ell - V_\ell V_{\ell-1}^T \right\|_F^2 \leq \frac{c_1 - 2 \log |Jf_\theta(x)|_+}{pL}.$$

Note how we not only obtain finite depth rates, but our result has the advantage of being applicable to any parameters with a sufficiently small parameter norm (close to the minimal norm solution). The bound is tighter at optimal parameters in which case $c_1 = R^{(1)}(f_\theta)$, but the theorem shows that the Bottleneck structure generalizes to points that are only almost optimal.

To prove that the pre-activations $\tilde{\alpha}_\ell(X)$ are approximately $k$-dimensional for some dataset $X$ (that may or may not be the training set) we simply need to show that the activations $\alpha_{\ell-1}(X)$ do not diverge, since $\tilde{\alpha}_\ell(X) = W_\ell \alpha_{\ell-1}(X) + b_\ell$ (and one can show that the bias will be small at almost every layer too). By our counterexample we know that we cannot rule out such explosion in general, however if we assume that the NTK [JGH18] $\Theta^{(L)}(x, x)$ is of order $O(L)$, then we can guarantee to convergence of the activations $\alpha_{\ell-1}(X)$ at almost every layer:

**Theorem 7.** *Given balanced parameters $\theta$ of a depth $L$ network, with $\|\theta\|^2 \leq kL + c_1$ and a point $x$ such that $\operatorname{Rank} J f_\theta(x) = k$ then if $\operatorname{Tr}\left[\Theta^{(L)}(x,x)\right] \leq cL$, then $\sum_{\ell=1}^{L} \|\alpha_{\ell-1}(x)\|_2^2 \leq \frac{c \max\{1, e^{\frac{c_1}{k}}\}}{k |J f_\theta(x)|_+^{2/k}} L$ and thus for all $p \in (0,1)$ there are at least $(1-p)L$ layers such that*

$$\|\alpha_{\ell-1}(x)\|_2^2 \leq \frac{1}{p} \frac{c \max\{1, e^{\frac{c_1}{k}}\}}{k |J f_\theta(x)|_+^{2/k}}.$$

Note that the balancedness assumption is not strictly necessary and could easily be weakened to some form of approximate balancedness, since we only require the fact that the parameter norm $\|W_\ell\|_F^2$ is well spread out throughout the layers, which follows from balancedness.

The NTK describes the narrowness of the minima [JGH20], and the assumption of bounded NTK is thus related to stability under large learning rates. There are multiple notions of narrowness that have been considered:

- The operator norm of the Hessian $H$ (which is closely related to the top eigenvalue of the NTK Gram matrix $\Theta^{(L)}(X,X)$ especially in the MSE case where at any interpolating function $\|H\|_{op} = \frac{1}{N}\left\|\Theta^{(L)}(X,X)\right\|_{op}$) which needs to be bounded by $2/\eta$ to have convergence when training with gradient descent with learning rate $\eta$.

- The trace of the Hessian (in the MSE case $\operatorname{Tr} H = \frac{1}{N}\operatorname{Tr}\Theta^{(L)}(X,X)$) which has been shown to describe the bias of stochastic gradient descent or approximation thereof [DML21, LWA21].

Thus boundedness of almost all activations as $L \to \infty$ can be guaranteed by assuming either $\frac{1}{N}\left\|\Theta^{(L)}(X,X)\right\|_{op} \leq cL$ (which implies $d_{out}\operatorname{Tr}\Theta^{(L)}(X,X) \leq cL$) or $\frac{1}{N}\operatorname{Tr}\Theta^{(L)}(X,X) \leq cL$ directly, corresponding to either gradient descent with $\eta = 2/cL$ or stochastic gradient descent with a similar scaling of $\eta$).

Note that one can find parameters that learn a function with a NTK that scales linearly in depth, but it is not possible to represent non-trivial functions with a smaller NTK $\Theta^{(L)} \ll L$. This is why we consider a linear scaling in depth to be the 'normal' size of the NTK, and anything larger to be narrow.

Putting the two Theorems together, we can prove the Bottleneck structure for almost all representations $\tilde{\alpha}_\ell(X)$:

**Corollary 8.** *Given balanced parameters $\theta$ of a depth $L$ network with $\|\theta\|^2 \leq kL + c_1$ and a set of points $x_1, \ldots, x_N$ such that $\operatorname{Rank} J f_\theta(x_i) = k$ and $\frac{1}{N}\operatorname{Tr}\left[\Theta^{(L)}(X,X)\right] \leq cL$, then for all $p \in (0,1)$ there are at least $(1-p)L$ layers such that*

$$s_{k+1}\left(\frac{1}{\sqrt{N}}\tilde{\alpha}_\ell(X)\right) \leq \sqrt{c_1 - 2\log|J f_\theta(x)|_+} \left(\sqrt{\frac{1}{N}\sum_{i=1}^{N}\frac{c \max\{1, e^{\frac{c_1}{k}}\}}{k |J f_\theta(x_i)|_+^{2/k}}} + \sqrt{p}\right)\frac{1}{p\sqrt{L}}.$$

### 4.1 Narrowness of rank-underestimating minima

We know that the large $R^{(1)}$ value of BN-rank 1 fitting functions is related to the explosion of its derivative, but a large Jacobian also leads to a blow up of the NTK:

**Proposition 9.** *For any point $x$, we have*

$$\left\|\partial_{xy}^2 \Theta(x,x)\right\|_{op} \geq 2L \|J f_\theta(x)\|_{op}^{2-2/L}$$

*where $\partial_{xy}^2 \Theta(x,x)$ is understood as a $d_{in}d_{out} \times d_{in}d_{out}$ matrix.*

*Furthermore, for any two points $x, y$ such that the pre-activations of all neurons of the network remain constant on the segment $[x,y]$, then either $\|\Theta(x,x)\|_{op}$ or $\|\Theta(y,y)\|_{op}$ is lower bounded by $\frac{L}{4}\|x-y\|^2 \left\|J f_\theta(x)\frac{y-x}{\|x-y\|}\right\|_2^{2-2/L}$.*

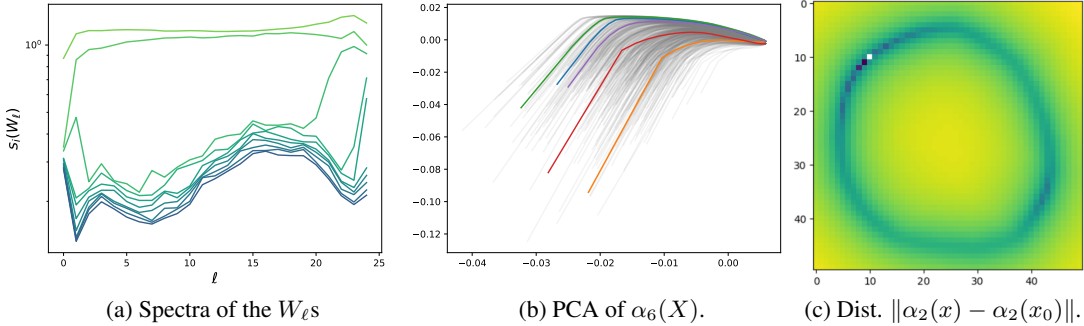

(a) Spectra of the $W_\ell$s      (b) PCA of $\alpha_6(X)$.      (c) Dist. $\|\alpha_2(x) - \alpha_2(x_0)\|$.

Figure 2: A depth $L = 25$ network with a width of 200 trained on the task described in Section 5 with a ridge $\lambda = 0.0002$. (a) Singular values of the weight matrices of the network, showing two outliers in the bottleneck, which implies that the network has recovered the true rank of 2. (b) Hidden representation of the 6-th layer projected to the first two dimensions, we see how images of GD paths do not cross in this space, showing that the dynamics on these two dimensions are self-consistent. (c) The distance $\|\alpha_2(x_0) - \alpha_2(x)\|$ in the second hidden layer between the representations at a fixed point $x_0$ (at the white pixel) and another point $x$ on a plane orthogonal to the axis $w$ of rotation, we see that all points on the same symmetry orbit are collapsed together, proving that the network has learned the rotation symmetry.

With some additional work, we can show the the NTK of such rank-underestimating functions will blow up, suggesting a narrow minimum:

**Theorem 10.** *Let $f^* : \Omega \to \mathbb{R}^{d_{out}}$ be a function with Jacobian rank $k^* > 1$ (i.e. there is a $x \in \Omega$ with $\mathrm{Rank} J f^*(x) = k^*$), then with high probability over the sampling of a training set $x_1, \ldots, x_N$ (sampled from a distribution with support $\Omega$), we have that for any parameters $\theta$ of a deep enough network that represent a BN-rank 1 function $f_\theta$ that fits the training set $f_\theta(x_i) = f^*(x_i)$ with norm $\|\theta\|^2 = L + c_1$ then there is a point $x \in \Omega$ where the NTK satisfies*

$$\mathrm{Tr}\left[\Theta^{(L)}(x,x)\right] \geq c'' L e^{-c_1} N^{4 - \frac{4}{k^*}}.$$

One the one hand, we know that $c_1$ must satisfy $c_1 \geq R^{(1)}(f_\theta) \geq 2(1 - \frac{1}{k^*}) \log N$ but if $c_1$ is within a factor of 2 of this lower bound $c_1 < 4(1 - \frac{1}{k^*}) \log N$, then the above shows that the NTK will blow up a rate $N^\alpha L$ for a positive $\alpha$.

The previous explanation for why GD avoids BN-rank 1 fitting functions was that when $N$ is much larger than the depth $L$ (exponentially larger), there is a rank-recovering function with a lower parameter norm than any rank-underestimating functions. But this relies on the assumption that GD converges to the lower norm minima, and it is only true for sufficiently small depths. In contrast the narrowness argument applies for any large enough depth and does not assume global convergence.

Of course the complete explanation might be a mix of these two reasons and possbily some other phenomenon too. Proving why GD avoids minima that underestimate the rank with a rank $1 < k < k^*$ also remains an open question.

## 5 Numerical Experiment: Symmetry Learning

In general, functions with a lot of symmetries have low BN-rank since a function $f$ with symmetry group $G$ can be decomposed as mapping the inputs $\Omega$ to the inputs module symmetries $\Omega/G$ and then mapping it to the outputs, thus $\mathrm{Rank}_{BN}(f; \Omega) \leq \dim \Omega/G$ where $\dim \Omega/G$ is the smallest dimension $\Omega/G$ can be embedded into. Thus the bias of DNNs to learn function with a low BN-rank can be interpreted as the network learning symmetries of the task. With this interpretation, overestimating the rank corresponds to failing to learn all symmetries of the task, while underestimating the rank can be interpreted as the network learning spurious symmetries that are not actual symmetries of the task.

To test this idea, we train a network to predict high dimensional dynamics with high dimensional symmetries. Consider the loss $C(v) = \left\| vv^T - \left( ww^T + E \right) \right\|_F^2$ where $w \in \mathbb{R}^d$ is a fixed unit vector and $E$ is a small noise $d \times d$ matrix. We optimize $v$ with gradient descent to try and fit the true vector $w$ (up to a sign). One can think of these dynamics as learning a shallow linear network $vv^T$ with a single hidden neuron. We will train a network to predict the evolution of the cost in time $C(v(t))$.

For small noise matrix $E$, the GD dynamics of $v(t)$ are invariant under rotation around the vector $w$. As a result, the high-dimensional dynamics of $v(t)$ can captured by only two *summary statistics* $u(v) = ((w^T v)^2, \left\| (I - ww^T)v \right\|^2)$: the first measures the position along the axis formed by $w$ and the second the distance to this axis [AGJ22]. The evolution of the summary statistics is (approximately) self-consistent (using the fact that $\|v\|^2 = (w^T v)^2 + \left\| (I - ww^T)v \right\|^2$):

$$\partial_t (w^T v)^2 = -8(\|v\|^2 - 1)(w^T v)^2 + O(\|E\|)$$
$$\partial_t \left\| (I - ww^T)v \right\|^2 = -8 \|v\|^2 \left\| (I - ww^T)v \right\|^2 + O(\|E\|).$$

Our goal now is to see whether a DNN can learn these summary statistics, or equivalently learn the underlying rotation symmetry. To test this, we train a network on the following supervised learning problem: given the vector $v(0)$ at initialization, predict the loss $(C(v(1)), \ldots, C(v(T)))$ over the next $T$ GD steps. For $E = 0$, the function $f^* : \mathbb{R}^d \to \mathbb{R}^T$ that is to be learned has BN-rank 2, since one can first map $v(0)$ to the corresponding summary statistics $u(v(0)) \in \mathbb{R}^2$, and then solve the differential equation on the summary statistics $(u(1), \ldots, u(T))$ over the next $T$ steps, and compute the cost $C(v) = \|v\|^4 - 2(w^T v)^2 + 1 + O(\|E\|)$ from $u$.

We observe in Figure 2 that a large depth $L_2$-regularized network trained on this task learns the rotation symmetry of the task and learns two dimensional hidden representations that are summary statistics (summary statistics are only defined up to bijections, so the learned representation match $u(v)$ only up to bijection but they can be recognized from the fact that the GF paths do not cross on the 2D representation).

## 6 Conclusion

We have computed corrections to the infinite depth description of the representation cost of DNNs given in [Jac23], revealing two regularity $R^{(1)}, R^{(2)}$ measures that balance against the dominating low rank/dimension bias $R^{(0)}$. We have also partially described another regularity inducing bias that results from large learning rates. We argued that these regularity bias play a role in stopping the network from underestimating the 'true' BN-rank of the task (or equivalently overfitting symmetries).

We have also proven the existence of a bottleneck structure in the weight matrices and under the condition of a bounded NTK of the learned representations, where most hidden representations are approximately $k = R^{(0)}(f_\theta)$-dimensional, with only a few high-dimensional representations.

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

# A Properties of the Corrections

## A.1 First Correction

**Theorem 11** (Theorem 3 from the main). *For all inputs $x$ where $\mathrm{Rank}Jf(x) = R^{(0)}(f;\Omega)$, we have $R^{(1)}(f) \geq 2\log|Jf(x)|_+$, furthermore:*

1. *If $R^{(0)}(f \circ g) = R^{(0)}(f) = R^{(0)}(g)$, then $R^{(1)}(f \circ g) \leq R^{(1)}(f) + R^{(1)}(g)$.*

2. *If $R^{(0)}(f + g) = R^{(0)}(f) + R^{(0)}(g)$, then $R^{(1)}(f + g) \leq R^{(1)}(f) + R^{(1)}(g)$.*

3. *If $P_{\mathrm{Im}A^T}\Omega$ and $A\Omega$ are $k = \mathrm{Rank}A$ dimensional and completely positive (i.e. they can be embedded with an isometric linear map into $\mathbb{R}_+^m$ for some $m$), then $R^{(1)}(x \mapsto Ax;\Omega) = 2\log|A|_+$ .*

*Proof.* For the first bound, we remember that $R(f;\Omega, L) \geq L\,\|Jf\|_{2/L}^{2/L}$, therefore

$$R^{(1)}(f;\Omega) = \lim_{L\to\infty} R(f;\Omega, L) - LR^{(0)}(f;\Omega)$$

$$\geq \lim_{L\to\infty} L \sum_{i=1}^{\mathrm{Rank}Jf(x)} s_i(Jf(x))^{\frac{2}{L}} - 1$$

$$\geq \sum_{i=1}^{\mathrm{Rank}Jf(x)} 2\log s_i(Jf(x))$$

where we used $s^{\frac{2}{L}} - 1 = e^{\frac{2}{L}\log s} - 1 \geq \frac{2}{L}\log s$.

(1) Since $R(f \circ g;\Omega, L_1 + L_2) \leq R(f;L_1) + R(g;L_2)$, we have

$$R^{(1)}(f \circ g;\Omega) = \lim_{L_1 + L_2 \to \infty} R(f \circ g;\Omega, L_1 + L_2) - (L_1 + L_2)R^{(0)}(f \circ g;\Omega)$$

$$\leq \lim_{L_1 \to \infty} R(f;\Omega, L_1) - L_1 R^{(0)}(f;\Omega) + \lim_{L_2 \to \infty} R(g;\Omega, L_2) - L_2 R^{(0)}(f;\Omega)$$

$$= R^{(1)}(f;\Omega) + R^{(1)}(g;\Omega).$$

(2) Since $R(f + g;\Omega, L) \leq R(f;\Omega, L) + R(g;\Omega, L)$, we have

$$R^{(1)}(f + g;\Omega) = \lim_{L\to\infty} R(f + g;\Omega, L) - LR^{(0)}(f + g;\Omega)$$

$$\leq \lim_{L\to\infty} R(f;\Omega, L) - LR^{(0)}(f;\Omega) + \lim_{L\to\infty} R(g;\Omega, L) - LR^{(0)}(g;\Omega)$$

$$= R^{(1)}(f;\Omega) + R^{(1)}(g;\Omega).$$

(3) By the first bound, we know that $R^{(1)}(x \mapsto Ax;\Omega) \geq 2\log|A|_+$, we now need to show $R^{(1)}(x \mapsto Ax;\Omega) \leq 2\log|A|_+$. Let us define the set of completely positive representations as the set of bilinear kernels $K(x,y) = x^T B^T By$ such that $Bx$ has non-negative entries for all $x \in \Omega$ (we say that a kernel $K$ is completely positive over $\Omega$ if it can be represented in this way for some choice of $B$). The set of completely positive representations is convex, since for $K(x,y) = x^T B^T By$ and $\tilde{K}(x,y) = x^T \tilde{B}^T \tilde{B}y$, we have

$$\frac{K(x,y) + \tilde{K}(x,y)}{2} = x^T \begin{pmatrix} \frac{1}{\sqrt{2}}B \\ \frac{1}{\sqrt{2}}\tilde{B} \end{pmatrix}^T \begin{pmatrix} \frac{1}{\sqrt{2}}B \\ \frac{1}{\sqrt{2}}\tilde{B} \end{pmatrix} y.$$

The conditions that there are $O_{in}$ and $O_{out}$ with $O_{in}^T O_{in} = P_{\mathrm{Im}A^T}$ and $O_{out}^T O_{out} = P_{\mathrm{Im}A}$ such that $O_{in}\Omega \in \mathbb{R}_+^{k_1}$ and $O_{out}A\Omega \in \mathbb{R}_+^{k_2}$ is equivalent to saying that the kernels $K_{in}(x,y) = x^T P_{\mathrm{Im}A^T} y$ and $K_{out}(x,y) = x^T A^T Ax$ are completely positive over $\Omega$.

By the convexity of completely positive representations, the interpolation $K_p = pK_{in} + (1-p)K_{out}$ is completely positive for all $p \in [0, 1]$. Now choose for all depths $L$ and all layers $\ell = 1, \ldots, L-1$

a matrix $B_{L,\ell}$ such that $K_{p=\frac{\ell}{L}}(x,y) = x^T B_{L,\ell}^T B_{L,\ell} y$ and then choose the weights $W_\ell$ of the depth $L$ network as

$$W_\ell = B_{L,\ell} B_{L,\ell-1}^+,$$

using the convention $B_{L,0} = I_{d_{in}}$ and $B_{L,L} = I_{out}$. By induction, we show that for any input $x \in \Omega$ the activation of the $\ell$-th hidden layer is $B_{L,\ell} x$. This is true for $\ell = 1$, since $W_1 = B_{L,1}$ and therefore $p^{(1)}(x) = B_{L,1} x$ which has positive entries so that $q^{(1)}(x) = \sigma\left(p^{(1)}(x)\right) = B_{L,1} x$. Then by induction

$$p^{(\ell)}(x) = W_\ell q^{(\ell-1)}(x) = B_{L,\ell} B_{L,\ell-1}^+ B_{L,\ell-1} x = B_{L,\ell} x,$$

which has positive entries, so that again $q^{(\ell)}(x) = \sigma\left(p^{(\ell)}(x)\right) = B_{L,\ell} x$. In the end, we get $p^{(L)}(x) = Ax$ as needed.

Let us now compute the Frobenius norms of the weight matrices $\|W_\ell\|_F^2 = \mathrm{Tr}\left[B_{L,\ell}^T B_{L,\ell}\left(B_{L,\ell-1}^T B_{L,\ell-1}\right)^+\right]$ as $L \to \infty$, remember that $B_{L,\ell}^T B_{L,\ell} = \frac{\ell}{L} P_{\mathrm{Im}A^T} + (1 - \frac{\ell}{L}) A^T A$, therefore the matrices $B_{L,\ell}^T B_{L,\ell}$ and $B_{L,\ell-1}^T B_{L,\ell-1}$ converge to each other, so that at first order $B_{L,\ell}^T B_{L,\ell}\left(B_{L,\ell-1}^T B_{L,\ell-1}\right)^+$ converges to $P_{\mathrm{Im}A^T}$, so that $\|W_\ell\|_F^2 \to \mathrm{Rank}A$, so that $\sum_{\ell=1}^L \|W_\ell\|_F^2 - L\mathrm{Rank}A$ converges to a finite value as $L \to \infty$. To obtain this finite limit, we need to study approximate the next order

$$\|W_\ell\|_F^2 - \mathrm{Rank}A = \sum_{i=1}^{\mathrm{Rank}A} 2\log s_i(W_i) + O(L^{-2})$$

$$= \log\left|B_{L,\ell}^T B_{L,\ell}\left(B_{L,\ell-1}^T B_{L,\ell-1}\right)^+\right|_+ + O(L^{-2})$$

$$= \log\left|B_{L,\ell}^T B_{L,\ell}\right|_+ - \log\left|B_{L,\ell-1}^T B_{L,\ell-1}\right|_+ + O(L^{-2}).$$

But as we sum all these second order terms, they cancel out, and we are left with

$$\sum_{\ell=1}^L \|W_\ell\|_F^2 - L\mathrm{Rank}A = 2\log|A|_+ - 2\log|I_{\mathrm{Im}A^T}|_+ + O(L^{-1}).$$

We have therefore build parameters $\theta$ that represent the function $x \mapsto Ax$ with parameter norm $\|\theta\|^2 = L\mathrm{Rank}A + 2\log|A|_+ + O(L^{-1})$, which upper bounds the representation cost, thus implying that $R^{(1)}(x \mapsto Ax; \Omega) \le 2\log|A|_+$ as needed. $\qquad\square$

## A.2 Identity

**Proposition 12** (Proposition 5 from the main). *For a domain with $\mathrm{Rank}_J(id; \Omega) = \mathrm{Rank}_{BN}(id; \Omega) = k$, then $R^{(1)}(id; \Omega) = 0$ if and only if $\Omega$ is $k$-planar and completely positive.*

*Proof.* First if $\Omega$ is completely positive and $k$-planar one can represent the identity with a depth $L$ network of parameter norm $Lk$, by taking $W_1 = O, W_\ell = P_{\mathrm{Im}O}, W_L = O^T$ where $O$ is the $m \times d$ so that $O\Omega \subset \mathbb{R}_+^m$ and $O^T O = P_{\mathrm{span}\Omega}$. Thus $R^{(1)}(id; \Omega) = 0$ (and all other corrections as well).

We will show that for any two points $x, y \in \Omega$ with $k$-dim tangent spaces, their tangent spaces must match if $R^{(1)}(id; \Omega) = 0$.

Let $A = J\alpha^{(L-1)}(x)_{|T_x\Omega}$ and $B = J\alpha^{(L-1)}(y)_{|T_y\Omega}$ be the be the Jacobian of the last hidden activations restricted to the tangent spaces, we know that

$$P_{T_x\Omega} = W_L A$$
$$P_{T_y\Omega} = W_L B$$

so that given any weight matrix $W_L$ whose image contains $T_x\Omega$ and $T_y\Omega$, we can write

$$A = W_L^+ P_{T_x\Omega}$$
$$B = W_L^+ P_{T_y\Omega}.$$

Without loss of generality, we may assume that the span of $T_x\Omega$ and $T_y\Omega$ is full output space, and therefore that $W_L W_L^T$ is invertible.

Now we now that any parameters that represent the identity on $\Omega$ and has $A = J\alpha^{(L-1)}(x)_{|T_x\Omega}$ and $B = J\alpha^{(L-1)}(y)_{|T_y\Omega}$ must have parameter norm at least

$$\|W_L\|_F^2 + k(L-1) + \max\left\{2\log|A|_+, 2\log|B|_+\right\}.$$

Subtracting $kL$ and taking $L \to \infty$, we obtain that

$$R^{(1)}(id;\Omega) \geq \min_{W_L} \|W_L\|_F^2 - k + \max\left\{2\log\left|W_L^+ P_{T_x\Omega}\right|_+, 2\log\left|W_L^+ P_{T_y\Omega}\right|_+\right\}.$$

If we optimize $W_L$ only up to scaling (i.e. optimize $aW_L$ over $a$) we see that at the optimum, we always have $\|W_L\|_F^2 = k$. This allows us to rewrite the optimization as

$$R^{(1)}(id;\Omega) \geq \min_{\|W_L\|_F^2 = k,} \max\left\{2\log\left|W_L^+ P_{T_x\Omega}\right|_+, 2\log\left|W_L^+ P_{T_y\Omega}\right|_+\right\}.$$

The only way to put the first term inside the maximum to 0 is to have $W_L W_L^T = P_{T_x\Omega}$, but this leads to an exploding second term if $P_{T_x\Omega} \neq P_{T_y\Omega}$. $\qquad\square$

Under the assumption of $C$-uniform Lipschitzness of the representations (that for all $\ell$, the functions $\tilde{\alpha}_\ell$ and $(\alpha_\ell \to f_\theta)$ are $C$-Lipschitz), one can show a stronger version of the above:

**Proposition 13.** *For a $C$-uniformly Lipschitz sequence of ReLU networks representing the function $f$, we have*

$$R^{(1)}(f) \geq \log|Jf(x)|_+ + \log|Jf(y)|_+ + C^{-2}\|Jf_\theta(x) - Jf_\theta(y)\|_*.$$

*Proof.* The decomposition of the difference

$$Jf_\theta(x) - Jf_\theta(y) = \sum_{\ell=1}^{L-1} W_L D_{L-1}(y)\cdots W_{\ell+1}\left(D_\ell(x) - D_\ell(y)\right)W_\ell D_{\ell-1}(x)\cdots D_1(x)W_1,$$

for the $w_\ell \times w_\ell$ diagonal matrices $D_\ell(x) = \mathrm{diag}(\dot{\sigma}(\tilde{\alpha}_\ell(x)))$, implies the bound

$$\|Jf_\theta(x) - Jf_\theta(y)\|_* \leq \sum_{\ell=1}^{L-1} \|W_L D_{L-1}(y)\cdots D_{\ell+1}(y)\|_{op} \|W_{\ell+1}\left(D_\ell(x) - D_\ell(y)\right)W_\ell\|_* \|D_{\ell-1}(x)\cdots D_1(x)W_1\|_{op}$$

$$\leq \frac{C^2}{2}\sum_{\ell=1}^{L-1}\left(\|W_{\ell+1}\left(D_\ell(x) - D_\ell(y)\right)\|_F^2 + \|\left(D_\ell(x) - D_\ell(y)\right)W_\ell\|_F^2\right)$$

since $\|AB\|_* \leq \frac{\|A\|_F^2 + \|B\|_F^2}{2}$ and $(D_\ell(x) - D_\ell(y))^2 = (D_\ell(x) - D_\ell(y))$.

Now since

$$L\|Jf_\theta(x)\|_{2/L}^{2/L} \leq \sum_{\ell=1}^{L} \|W_\ell D_{\ell-1}(x)\|_F^2$$

$$L\|Jf_\theta(x)\|_{2/L}^{2/L} \leq \sum_{\ell=1}^{L} \|D_\ell(x)W_\ell\|_F^2$$

with the convention $D_0(x) = I_{d_{in}}$ and $D_L(x) = I_{d_{out}}$. We obtain that

$$L\|Jf_\theta(x)\|_{2/L}^{2/L} + L\|Jf_\theta(y)\|_{2/L}^{2/L} \leq \frac{1}{2}\sum_{\ell=1}^{L}\|W_\ell D_{\ell-1}(x)\|_F^2 + \|W_\ell D_{\ell-1}(y)\|_F^2 + \|D_\ell(x)W_\ell\|_F^2 + \|D_\ell(y)W_\ell\|_F^2$$

$$\leq \sum_{\ell=1}^{L} 2\|W_\ell\|_F^2 - \frac{1}{2}\|W_\ell\left(D_{\ell-1}(x) - D_{\ell-1}(y)\right)\|_F^2 - \frac{1}{2}\|\left(D_\ell(x) - D_\ell(y)\right)W_\ell\|_F^2.$$

This implies the bound

$$\|\theta\|^2 \geq \frac{L\, \|Jf_\theta(x)\|_{2/L}^{2/L} + L\, \|Jf_\theta(y)\|_{2/L}^{2/L}}{2} + C^{-2}\, \|Jf_\theta(x) - Jf_\theta(y)\|_*$$

and thus

$$R^{(1)}(f) \geq \log|Jf(x)|_+ + \log|Jf(y)|_+ + C^{-2}\, \|Jf_\theta(x) - Jf_\theta(y)\|_* .$$

$\square$

### A.3 Second Correction

**Proposition 14** (Proposition 4 from the main). *If there is a limiting representation as $L \to 0$ in the optimal representation of $f$, then $R^{(2)}(f) \geq 0$. Furthermore:*

1. *If $R^{(0)}(f \circ g) = R^{(0)}(f) = R^{(0)}(g)$ and $R^{(1)}(f \circ g) = R^{(1)}(f) + R^{(1)}(g)$, then $\sqrt{R^{(2)}(f \circ g)} \leq \sqrt{R^{(2)}(f)} + \sqrt{R^{(2)}(g)}$.*

2. *If $R^{(0)}(f + g) = R^{(0)}(f) + R^{(0)}(g)$ and $R^{(1)}(f + g) = R^{(1)}(f) + R^{(1)}(g)$, then $R^{(2)}(f + g) \leq R^{(2)}(f) + R^{(2)}(g)$.*

3. *If $A^p \Omega$ is $k = \mathrm{Rank} A$-dimensional and completely positive for all $p \in [0,1]$, where $A^p$ has its non-zero singular taken to the $p$-th power, then $R^{(2)}(x \mapsto Ax; \Omega) = \frac{1}{2} \left\| \log_+ A^T A \right\|^2$.*

*Proof.* We start from the inequality

$$R(f \circ g; \Omega, L_f + L_g) \leq R(f; g(\Omega), L_f) + R(g; \Omega, L_g).$$

We subtract $(L_f + L_g)R^{(0)}(f \circ g) + R^{(1)}(f \circ g)$ divide by $L_f + L_g$ and take the limit of increasing depths $L_f, L_g$ with $\lim_{L_g, L_f \to \infty} \frac{L_f}{L_f + L_g} = p \in (0,1)$ to obtain

$$R^{(2)}(f \circ g; \Omega) \leq \frac{1}{1-p} R^{(2)}(f; g(\Omega)) + \frac{1}{p} R^{(2)}(g; \Omega). \tag{1}$$

If $K_p$ is the limiting representation at a ratio $p \in (0,1)$, we have $R^{(2)}(f; \Omega) = \frac{1}{p} R^{(2)}(K_p; \Omega) + \frac{1}{1-p} R^{(2)}(K_p \to f; \Omega)$ and $p$ must minimize the RHS since if it was instead minimized at a different ratio $p' \neq p$, one could find a lower norm representation by mapping to $K_p$ in the first $p'L$ layers and then back to the outputs. Now there are two possiblities, either $R^{(2)}(K_p; \Omega)$ and $R^{(2)}(K_p \to f; \Omega)$ are non-negative in which case the minimum is attained at some $p \in (0,1)$ and $R^{(2)}(f; \Omega) \geq 0$, or one or both is negative in which case the above is minimized at $p \in \{0, 1\}$ and $R^{(2)}(f; \Omega) = -\infty$. Since we assumed $p \in (0,1)$, we are in the first case.

(1) To prove the first property, we optimize the RHS of 1 over all possible choices of $p$ (and assuming that $R^{(2)}(f; g(\Omega)), R^{(2)}(g; \Omega) \geq 0$) we obtain

$$\sqrt{R^{(2)}(f \circ g; \Omega)} \leq \sqrt{R^{(2)}(f; g(\Omega))} + \sqrt{R^{(2)}(g; \Omega)}.$$

(2) This follows from the inequality $R(f + g; \Omega, L) \leq R(f; g(\Omega), L) + R(g; \Omega, L)$ after subtracting the $R^{(0)}$ and $R^{(1)}$ terms, dividing by $L$ and taking $L \to \infty$.

(3) If $A = USV^T$, one chooses $W_\ell = U_\ell S^{\frac{1}{L}} U_{\ell-1}^T$ with $U_0 = V$, $U_L = U$ and $U_\ell$ chosen so that $U_\ell S^{\frac{\ell}{L}} V^T \Omega \in \mathbb{R}_+^{n_\ell}$, choosing large enough widths $n_\ell$. This choice of representation of $A$ is optimal, i.e. its parameter norm matches the representation cost $L\mathrm{Tr}\left[S^{\frac{2}{L}}\right] = L\mathrm{Rank}A + 2\log|A|_+ + \frac{1}{2L}\left\|\log_+ A^T A\right\|^2 + O(L^{-2})$.

We know that

$$\lim_{L \to \infty} R^{(1)}(\alpha_{\ell_1} \to \alpha_{\ell_2}; \Omega) = R^{(1)}(f_\theta; \Omega) \lim_{L \to \infty} \frac{\ell_2 - \ell_1}{L}$$

$$\frac{1}{p}R^{(2)}(\alpha;\Omega) + \frac{1}{1-p}R^{(2)}(\alpha \to f;\Omega) \geq R^{(2)}(f;\Omega)$$

$$\frac{1}{p}R^{(2)}(\alpha;\Omega) + \frac{1}{1-p}R^{(2)}(\alpha \to f;\Omega) \geq R^{(2)}(f;\Omega)$$

□

## B Bottleneck Structure

This first result shows the existence of a Bottleneck structure on the weight matrices:

**Theorem 15** (Theorem 6 from the main). *Given parameters $\theta$ of a depth $L$ network, with $\|\theta\|^2 \leq kL + c_1$ and a point $x$ such that $\mathrm{Rank}Jf_\theta(x) = k$, then there are $w_\ell \times k$ (semi-)orthonormal $U_\ell, V_\ell$ such that*

$$\sum_{\ell=1}^{L} \left\| W_\ell - U_\ell V_{\ell+1}^T \right\|_F^2 + \|b_\ell\|^2 \leq c_1 - 2\log|Jf_\theta(x)|_+$$

*thus for any $p \in (0,1)$ there are at least $(1-p)L$ layers $\ell$ with*

$$\left\| W_\ell - U_\ell V_{\ell+1}^T \right\|_F^2 + \|b_\ell\|^2 \leq \frac{c_1 - 2\log|Jf_\theta(x)|_+}{pL}$$

*Proof.* Since

$$
\begin{aligned}
Jf_\theta(x) &= W_L D_{L-1}(x) \cdots D_1(x) W_1 \\
&= W_L P_{\mathrm{Im}J\alpha_{L-1}(x)} D_{L-1}(x) \cdots P_{\mathrm{Im}J\alpha_1(x)} D_1(x) W_1
\end{aligned}
$$

If the preimage of $A$ matches the image of $B$ then $|AB|_+ = |A|_+ |B|_+$. We therefore have

$$
\begin{aligned}
\log|Jf_\theta(x)|_+ &= \log\left|W_L P_{\mathrm{Im}J\alpha_{L-1}(x)}\right|_+ \\
&\quad + \log\left|P_{\mathrm{Im}J(\alpha_{L-1}\to f_\theta)(x)^T} D_{L-1}(x) P_{\mathrm{Im}J\tilde\alpha_{L-1}(x)}\right|_+ + \log\left|P_{\mathrm{Im}J(\tilde\alpha_{L-1}\to f_\theta)(x)^T} W_{L-1} P_{\mathrm{Im}J\alpha_{L-2}(x)}\right|_+ \\
&\quad + \ldots \\
&\quad + \log\left|P_{\mathrm{Im}J(\alpha_1\to f_\theta)(x)^T} D_1(x) P_{\mathrm{Im}J\tilde\alpha_1(x)}\right|_+ + \log\left|P_{J(\tilde\alpha_1\to f_\theta)(x)^T} W_1\right|_+
\end{aligned}
$$

This implies that

$$
\begin{aligned}
&\sum_{\ell=1}^{L} \|W_\ell\|_F^2 - k - 2\log\left|P_{\mathrm{Im}J(\tilde\alpha_\ell\to f_\theta)(x)^T} W_\ell P_{\mathrm{Im}J\alpha_{\ell-1}(x)}\right|_+ \\
&\quad - 2\log\left|P_{\mathrm{Im}J(\alpha_\ell\to f_\theta)(x)^T} D_\ell(x) P_{\mathrm{Im}J\tilde\alpha_\ell(x)}\right| \\
&= \|\theta\|^2 - kL - 2\log|Jf_\theta(x)|_+ \\
&\leq c_1 - 2\log|Jf_\theta(x)|_+
\end{aligned}
$$

with the convention $D_L(x) = I_{w_{out}}$.

Our goal is to show that the LHS is a sum of positive value which sum up to a finite positive value, which will imply that most of the summands must be very small.

First observe that

$$-2\log\left|P_{\mathrm{Im}J(\alpha_\ell\to f_\theta)(x)^T} D_\ell(x) P_{\mathrm{Im}J\tilde\alpha_\ell(x)}\right| \geq k - \left\|P_{\mathrm{Im}J(\alpha_\ell\to f_\theta)(x)^T} D_\ell(x) P_{\mathrm{Im}J\tilde\alpha_\ell(x)}\right\|_F^2$$

which is positive since the eigenvalues of $D_\ell(x)$ are $\leq 1$.

To show that the other part of the summands $\|W_\ell\|_F^2 - k - 2\log\left|P_{\mathrm{Im}J(\tilde\alpha_\ell\to f_\theta)(x)^T} W_\ell P_{\mathrm{Im}J\alpha_{\ell-1}(x)}\right|_+$ is positive, we give lower bound it. First, it can be rewritten as

$$
\begin{aligned}
&\left(\left\|P_{\mathrm{Im}J(\tilde\alpha_\ell\to f_\theta)(x)^T} W_\ell P_{\mathrm{Im}J\alpha_{\ell-1}(x)}\right\|_F^2 - k - 2\log\left|P_{\mathrm{Im}J(\tilde\alpha_\ell\to f_\theta)(x)^T} W_\ell P_{\mathrm{Im}J\alpha_{\ell-1}(x)}\right|_+\right) \\
&\quad + \left\|W_\ell - P_{\mathrm{Im}J(\tilde\alpha_\ell\to f_\theta)(x)^T} W_\ell P_{\mathrm{Im}J\alpha_{\ell-1}(x)}\right\|_F^2
\end{aligned}
$$

Now for a general matrix $A$, we have

$$\|A\|_F^2 - \text{Rank}A - 2\log|A|_+ = \sum_{i=1}^{\text{Rank}A} s_i(A)^2 - 1 - 2\log s_i(A)$$

$$\geq \sum_{i=1}^{\text{Rank}A} s_i(A)^2 - 1 - 2(s_i(A) - 1)$$

$$= \sum_{i=1}^{\text{Rank}A} (s_i(A) - 1)^2$$

$$= \left\|A - UV^T\right\|_F^2$$

for the SVD decomposition $A = USV^T$. We can thus lower bound

$$\|W_\ell\|_F^2 - k - 2\log\left|P_{\text{Im}J(\tilde{\alpha}_\ell \to f_\theta)(x)^T}W_\ell P_{\text{Im}J\alpha_{\ell-1}(x)}\right|_+ \geq \left\|W_\ell - U_\ell V_\ell^T\right\|_F^2$$

where $U_\ell S_\ell V_\ell^T$ is the SVD decomposition of $P_{\text{Im}J(\tilde{\alpha}_\ell \to f_\theta)(x)^T}W_\ell P_{\text{Im}J\alpha_{\ell-1}(x)}$ (which we know has rank $k$ since it must match the rank of $Jf_\theta(x)$).

Since $\|\theta\|^2 = \sum_\ell \|W_\ell\|_F^2 + \|b_\ell\|^2 \leq kL + c_1$, we have

$$\sum_{\ell=1}^{L} \left\|W_\ell - U_\ell V_{\ell+1}^T\right\|_F^2 + \|b_\ell\|^2 \leq c_1 - 2\log|Jf_\theta(x)|_+ .$$

And for any $p \in (0,1)$ there are at most $pL$ layers $\ell$ with

$$\left\|W_\ell - U_\ell V_{\ell+1}^T\right\|_F^2 + \|b_\ell\|^2 \leq \frac{c_1 - 2\log|Jf_\theta(x)|_+}{pL} .$$

$\square$

The fact that almost all weight matrices $W_\ell$ are approximately $k$-dim would imply that the pre-activations $\tilde{\alpha}_\ell(X) = W_\ell \alpha_{\ell-1}(X)$ are $k$-dim too under the condition that the activations $\alpha_{\ell-1}(X)$ do not diverge. Assuming a bounded NTK is sufficient to guarantee that these activations converge at almost every layer:

**Theorem 16** (Theorem 7 from the main). *Given balanced parameters $\theta$ of a depth $L$ network, with $\|\theta\|^2 \leq kL + c_1$ and a point $x$ such that $\text{Rank}Jf_\theta(x) = k$ then if $\frac{1}{N}\text{Tr}\left[\Theta^{(L)}(x,x)\right] \leq cL$, then $\sum_{\ell=1}^{L}\|\alpha_{\ell-1}(x)\|_2^2 \leq \frac{c\max\{1, e^{\frac{c_1}{k}}\}}{k|Jf_\theta(x)|_+^{2/k}}L$ and thus for all $p \in (0,1)$ there are at least $(1-p)L$ layers such that*

$$\|\alpha_{\ell-1}(x)\|_2^2 \leq \frac{1}{p}\frac{c\max\{1, e^{\frac{c_1}{k}}\}}{k\,|Jf_\theta(x)|_+^{2/k}} .$$

*Proof.* We have

$$\text{Tr}\left[\Theta^{(L)}(x,x)\right] = \sum_{\ell=1}^{L}\|\alpha_{\ell-1}(x)\|_2^2\,\|J(\tilde{\alpha}_\ell \to \alpha_L)(x)\|_F^2 ,$$

we therefore need to lower bound $\|J(\tilde{\alpha}_\ell \to \alpha_L)(x)\|_F^2$ to show that the activations $\|\alpha_{\ell-1}(x)\|_2^2$ must be bounded at almost every layer.

We will lower bound $\|J(\tilde{\alpha}_\ell \to \alpha_L)(x)\|_F^2$ by $\|J(\tilde{\alpha}_\ell \to \alpha_L)(x)P_\ell\|_F^2$ for $P_\ell$ the orthogonal projection to the image $\text{Im}J\tilde{\alpha}_\ell(x)$. Note $J(\tilde{\alpha}_\ell \to \alpha_L)(x)P_\ell$ and $Jf_\theta(x)$ have the same rank.

By the arithmetic-geometric mean inequality, we have $\|A\|_F^2 \geq \text{Rank}A\,|A|_+^{2/k}$, yielding

$$\|J(\tilde{\alpha}_\ell \to \alpha_L)(x)P_\ell\|_F^2 \geq k\,|J(\tilde{\alpha}_\ell \to \alpha_L)(x)P_\ell|_+^{2/k} .$$

Now the balancedness of the parameters (i.e. $\|W_{\ell,i\cdot}\|^2 + b_{\ell,i}^2 = \|W_{\ell+1,\cdot i}\|^2, \forall \ell, i$) implies that the parameter norms are increasing $\|W_{\ell+1}\|_F^2 \geq \|W_\ell\|_F^2$ and thus

$$\frac{\|W_{\ell+1}\|^2 + \cdots + \|W_L\|^2}{L - \ell} \geq \frac{\|W_1\|^2 + \cdots + \|W_\ell\|^2}{\ell}.$$

Thus

$$\frac{\|W_1\|^2 + \cdots + \|W_\ell\|^2}{\ell} = \frac{\left(\|W_1\|^2 + \cdots + \|W_\ell\|^2\right)}{L} + \frac{L - \ell}{L}\frac{\left(\|W_1\|^2 + \cdots + \|W_\ell\|^2\right)}{\ell}$$

$$\leq \frac{\|\theta\|^2}{L}$$

and

$$|J\tilde{\alpha}_\ell(x)|_+^{2/k\ell} \leq \frac{1}{k}\|J\tilde{\alpha}_\ell(x)\|_{2/\ell}^{2/\ell} \leq \frac{\|W_1\|^2 + \cdots + \|W_\ell\|^2}{k\ell} \leq \frac{\|\theta\|^2}{kL} \leq 1 + \frac{c_1}{kL}$$

and therefore

$$\|J(\tilde{\alpha}_\ell \to f_\theta)(x)P_\ell\|_F^2 \geq k\,|J(\tilde{\alpha}_\ell \to f_\theta)(x)P_\ell|_+^{2/k}$$

$$= k\frac{|Jf_\theta(x)|_+^{2/k}}{|J\tilde{\alpha}(x)|_+^{2/k}}$$

$$\geq k\frac{|Jf_\theta(x)|_+^{2/k}}{\left(1 + \frac{c_1}{L}\right)^\ell}$$

$$\geq k\,|Jf_\theta(x)|_+^{2/k}\,e^{-\frac{\ell}{L}\frac{c_1}{k}}$$

$$\geq k\,|Jf_\theta(x)|_+^{2/k}\,\min\{1, e^{-\frac{c_1}{k}}\}.$$

Thus

$$\sum_{\ell=1}^L \|\alpha_{\ell-1}(x)\|_2^2 \leq \frac{c\max\{1, e^{\frac{c_1}{k}}\}}{k\,|Jf_\theta(x)|_+^{2/k}}L$$

which implies that there are at most $pL$ layers $\ell$ with

$$\|\alpha_{\ell-1}(x)\|_2^2 \geq \frac{1}{p}\frac{c\max\{1, e^{\frac{c_1}{k}}\}}{k\,|Jf_\theta(x)|_+^{2/k}}.$$

$\square$

Note that for the MSE loss in the limit $\lambda \searrow 0$, the Hessian at a global minimum (i.e. $f_\theta$ interpolates the training set) equals $\mathrm{Tr}\,[\mathcal{HL}(\theta)] = \frac{1}{N}\mathrm{Tr}\,[\Theta^{(L)}(X, X)]$. If we then assume that the trace of the Hessian is bounded by $cL$, we get that $\mathrm{Tr}\,[\Theta^{(L)}(X, X)] \leq cNL$ and thus there are at least $(1 - p)L$ layers where

$$\frac{1}{N}\|\alpha_{\ell-1}(X)\|_F^2 \leq \frac{1}{p}\frac{c\max\{1, e^{\frac{c_1}{k}}\}}{k\,|Jf_\theta(x)|_+^{2/k}},$$

thus guaranteeing the infinite depth convergence of training set activations $\alpha_{\ell-1}(X)$ on those layers.

Putting the two above theorems together, we can prove that the pre-activations are $k$-dim at almost every layer:

**Corollary 17** (Corollary 8 from the main)**.** *Given balanced parameters $\theta$ of a depth $L$ network with $\|\theta\|^2 \leq kL + c_1$ and a set of points $x_1, \ldots, x_N$ such that $\mathrm{Rank}Jf_\theta(x_i) = k$ and $\frac{1}{N}\mathrm{Tr}\,[\Theta^{(L)}(X, X)] \leq cL$, then for all $p \in (0, 1)$ there are at least $(1 - p)L$ layers such that*

$$s_{k+1}\left(\frac{1}{\sqrt{N}}\tilde{\alpha}_\ell(X)\right) \leq \sqrt{c_1 - 2\log|Jf_\theta(x)|_+}\left(\sqrt{\frac{1}{N}\sum_{i=1}^N \frac{c\max\{1, e^{\frac{c_1}{k}}\}}{k\,|Jf_\theta(x_i)|_+^{2/k}}} + \sqrt{p}\right)\frac{1}{p\sqrt{L}}$$

*Proof.* Since $\tilde{\alpha}_\ell(X) = W_\ell \alpha_{\ell-1}(X) + b_\ell \mathbf{1}_N^T$ we know that

$$s_{k+1}\left(\frac{1}{\sqrt{N}}\tilde{\alpha}_\ell(X)\right) \leq s_{k+1}(W_\ell)\left\|\frac{1}{\sqrt{N}}\alpha_{\ell-1}(X)\right\|_{op} + \|b_\ell\|.$$

By Theorems 15 and 16, there are for any $p \in (0, \frac{1}{2})$ at least $(1-2p)L$ layers such that

$$\left\|W_\ell - V_\ell V_{\ell-1}^T\right\|_F^2 + \|b_\ell\|^2 \leq \frac{c_1 - 2\log|Jf_\theta(x)|_+}{pL}$$

$$\left\|\frac{1}{\sqrt{N}}\alpha_{\ell-1}(X)\right\|_F^2 \leq \sum_{i=1}^N \frac{1}{p}\frac{c\max\{1, e^{\frac{c_1}{k}}\}}{k\,|Jf_\theta(x_i)|_+^{2/k}}$$

so that

$$s_{k+1}\left(\frac{1}{\sqrt{N}}\tilde{\alpha}_\ell(X)\right) \leq \sqrt{\frac{c_1 - 2\log|Jf_\theta(x)|_+}{p}}\left(\sqrt{\frac{1}{pN}\sum_{i=1}^N \frac{c\max\{1, e^{\frac{c_1}{k}}\}}{k\,|Jf_\theta(x_i)|_+^{2/k}}} + 1\right)\frac{1}{\sqrt{L}}.$$

$\square$

# C  Local Minima Stability

In this section we motivate the assumption that the Jacobian $J\tilde{\alpha}_\ell(x)$ is uniformly bounded in operator norm as $L \to \infty$. The idea is that solutions with a blowing up Jacobian $J\tilde{\alpha}_\ell(x)$ correspond to very narrow local minima.

The narrowness of a local minimum is related to the Neural Tangent Kernel (or Fisher matrix). We have that

$$\mathrm{Tr}\left[\Theta^{(L)}(x,x)\right] = \sum_{\ell=1}^L \|\alpha_{\ell-1}(x)\|^2 \|J(\tilde{\alpha}_\ell \to f_\theta)(x)\|_F^2.$$

A large Jacobian $Jf_\theta(x)$ leads to a blow up of the derivative of the NTK:

**Proposition 18** (Proposition 9 from the main). *For any point $x$, we have*

$$\left\|\partial_{xy}^2 \Theta(x,x)\right\|_{op} \geq 2L\,\|Jf_\theta(x)\|_{op}^{2-2/L}$$

*where $\partial_{xy}^2\Theta(x,x)$ is understood as a $d_{in}d_{out} \times d_{in}d_{out}$ matrix.*

*Furthermore, for any two points $x, y$ such that the pre-activations of all neurons of the network remain constant on the segment $[x, y]$, then either $\|\Theta(x,x)\|_{op}$ or $\|\Theta(y,y)\|_{op}$ is lower bounded by $\frac{L}{4}\|x - y\|^2\left\|Jf_\theta(x)\frac{y-x}{\|x-y\|}\right\|_2^{2-2/L}$.*

*Proof.* (1) For any point $x$, we have

$$\partial_{x,y}\left(v^T\Theta(x,x)v\right)[u,u] = \sum_{\ell=1}^L u^T W_1^T D_1(x)\cdots D_{\ell-1}(x)^2\cdots D_1(x)W_1 uv^T W_L D_{L-1}(x)\cdots D_\ell(x)^2\cdots D_{L-1}(x)W_L^T v$$

$$= \sum_{\ell=1}^L \|D_{\ell-1}(x)\cdots D_1(x)W_1 u\|_2^2 \|D_\ell(x)\cdots D_{L-1}(x)W_L v\|_2^2.$$

On the other hand, we have

$$\left|v^T Jf_\theta(x)u\right| = \left|v^T W_L D_{L-1}(x)\cdots D_1(x)W_1 u\right|$$
$$\leq \|D_\ell(x)\cdots D_1(x)W_1 u\|_2 \|D_\ell(x)\cdots D_{L-1}(x)W_L v\|_2,$$

where we used the fact that $D_\ell(x)D_\ell(x) = D_\ell(x)$. This applies to the case $\ell = L$ and $\ell = 1$ too, using the definition $D_L(x) = I_{d_{out}}$ and $D_0(x) = I_{d_{in}}$. This implies

$$\partial_{xy}^2 \left( v^T \Theta(x,x) v \right)[u,u] \geq \left| v^T J f_\theta(x) u \right|^2 \sum_{\ell=1}^{L} \frac{\| D_{\ell-1}(x) \cdots D_1(x) W_1 u \|_2^2}{\| D_\ell(x) \cdots D_1(x) W_1 u \|_2^2}$$

$$\geq \left| v^T J f_\theta(x) u \right|^2 L \left( \frac{\| u \|_2^2}{\| W_L D_{L-1}(x) \cdots D_1(x) W_1 u \|_2^2} \right)^{\frac{1}{L}}$$

$$\geq L \frac{\left| v^T J f_\theta(x) u \right|^2}{\| J f_\theta(x) u \|_2^{2/L}}.$$

where we used the geometric/arithmetic mean inequality for the second inequality.

If $u, v$ are right and left singular vectors of $J f_\theta(x)$ with singular value $s$, then the above bound equals $L s^{2 - \frac{2}{L}}$.

(2) Now let us consider a segment $\gamma(t) = (1-t)x + ty$ between two points $x, y$ with no changes of activations on these paths (i.e. $D_\ell(\gamma(t))$ is constant for all $t \in [0,1]$). Defining $u = \frac{y-x}{\|y-x\|}$ and $v = \frac{J f_\theta(x) u}{\| J f_\theta(x) u \|}$, we have

$$\partial_t v^T \Theta(\gamma(t), \gamma(t)) v = \| x - y \| \, \partial_x \left( v^T \Theta(\gamma(t), \gamma(t)) v \right)[u] + \| x - y \| \, \partial_y \left( v^T \Theta(\gamma(t), \gamma(t)) v \right)[u]$$

and since $\partial_{xx} \Theta(\gamma(t), \gamma(t)) = 0$ and $\partial_{yy} \Theta(\gamma(t), \gamma(t)) = 0$ for all $t \in [0,1]$, we have

$$\partial_t^2 \left( v^T \Theta(\gamma(t), \gamma(t)) v \right) = 2 \| x - y \|^2 \, \partial_{xy}^2 \left( v^T \Theta(\gamma(t), \gamma(t)) v \right)[u,u] \geq 2L \| x - y \|^2 \| J f_\theta(x) u \|_2^{2 - 2/L}.$$

Since $v^T \Theta(\gamma(t), \gamma(t)) v \geq 0$ for all $t \in [0,1]$ then either

$$v^T \Theta(x,x) v \geq \frac{L}{4} \| x - y \|^2 \| J f_\theta(x) u \|_2^{2 - 2/L}$$

or

$$v^T \Theta(y,y) v \geq \frac{L}{4} \| x - y \|^2 \| J f_\theta(x) u \|_2^{2 - 2/L}.$$

$\square$

Rank-underestimating fitting functions typically feature exploding derivatives, which was used to show in [Jac23] that BN-rank 1 fitting functions must have a $R^{(1)}$ term that blows up iwith the number of datapoints $N$. With some additional work, we can show that the NTK will blow up at some $x$:

**Theorem 19** (Theorem 10 from the main). *Let $f^* : \Omega \to \mathbb{R}^{d_{out}}$ be a function with Jacobian rank $k^* > 1$ (i.e. there is a $x \in \Omega$ with $\mathrm{Rank} J f^*(x) = k^*$), then with high probability over the sampling of a training set $x_1, \ldots, x_N$ (sampled from a distribution with support $\Omega$), we have that for any parameters $\theta$ of a deep enough network that represent a BN-rank 1 function $f_\theta$ that fits the training set $f_\theta(x_i) = f^*(x_i)$ with norm $\| \theta \|^2 = L + c_1$ then there is a point $x \in \Omega$ where the NTK satisfies*

$$\Theta^{(L)}(x,x) \geq c'' L e^{-c_1} N^{4 - \frac{4}{k^*}}.$$

*Proof.* For all $i$ we define $d_{1,i}$ and $d_{2,i}$ to be the distance between $y_i$ and its closest and second closest point in $y_1, \ldots, y_N$. Following the argument in [BHH59], the shortest path that goes through all points must be at least $\sum_{i=1}^{N} \frac{d_{1,i} + d_{2,i}}{2}$ (which would be tight if it is possible to always jump to the closest or second closest point along the path). Since the expected distances $d_{1,i}$ and $d_{2,i}$ are $N^{-\frac{1}{k^*}}$ since the $y_i$ are sampled from a $k^*$-dimensional distribution, the expected length of the shortest path is of order $N^{1 - \frac{1}{k^*}}$. Actually most of the distance $d_{1,i}$ and $d_{2,i}$ will be of order $N^{-\frac{1}{k^*}}$ with only a few outliers with larger or smaller distances, thus for any subset of indices $I \subset [1, \ldots, N]$ that contains a finite ratio of all indices, the sum $\sum_{i \in I} \frac{d_{1,i} + d_{2,i}}{2}$ will be of order $N^{1 - \frac{1}{k^*}}$ too.

Following the argument in the proof Theorem 2 from [Jac23], we can reorder the indices so that the segment $[x_1, x_N]$ will mapped $f_\theta$ to a path that goes through $y_1, \ldots, y_N$. We can therefore define the points $\tilde{x}_1, \ldots, \tilde{x}_N$ that are preimages of $y_1, \ldots, y_N$ on the segment.

On the interval $[\tilde{x}_i, \tilde{x}_{i+1}]$ there must a point $x$ with $\|Jf_\theta(x)\|_{op} \geq \frac{\|y_{i+1} - y_i\|}{\|\tilde{x}_{i+1} - \tilde{x}_i\|}$. Now since $\sum \|\tilde{x}_{i+1} - \tilde{x}_i\| = \|x_N - x_1\| \leq \text{diag}\Omega$ there must be at least $p(N-1)$ intervals $i$ with $\|\tilde{x}_{i+1} - \tilde{x}_i\| \leq \frac{\text{diag}\Omega}{(1-p)(N-1)}$, and amongst those $i$s they would all satisfy $\|y_{i+1} - y_i\| \geq cN^{-\frac{1}{k^*}}$ except for a few outliers. Thus we can for example guarantee that there are at least $\frac{5}{6}(N-1)$ intervals $[\tilde{x}_i, \tilde{x}_{i+1}]$ that contain a point $x$ with $\|Jf_\theta(x)\|_{op} \geq cN^{1-\frac{1}{k^*}}$.

First observe that by Theorem 2 of [Jac23], there must be a point $x \in \Omega$ such that $\|Jf_\theta(x)\|_{op} \geq N^{1-\frac{1}{k}}$ thus by Theorem 6, there are at least $pL$ layers where

$$\left\| W_\ell - u_\ell v_\ell^T \right\|^2 \leq \frac{c_1 - 2\log|Jf_\theta(x)|_+}{pL}.$$

Consider one of those $pL$ layers $\ell$, with activatons $z_i = \alpha_{\ell-1}(\tilde{x}_i)$. Let $i_1, \ldots, i_N$ be the ordering of the indices so that $u_\ell^T z_{i_m}$ is increasing in $m$. Then the hidden representations must satisfy

$$\frac{\left\| W_\ell(z_{i_m} - z_{i_{m-1}}) \right\| + \left\| W_\ell(z_{i_m} - z_{i_{m+1}}) \right\|}{2} \geq e^{-\frac{\left\| \theta^{(\ell+1:L)} \right\|^2 - (L-\ell)}{2}} \frac{d_{1,i_m} + d_{2,i_m}}{2}$$

and

$$\left\| W_\ell(z_{i_m} - z_{i_{m-1}}) \right\| \leq u_\ell^T(z_{i_m} - z_{i_{m-1}}) + \sqrt{\frac{c_1 - 2\log|Jf_\theta(x)|_+}{pL}} \left\| z_{i_m} - z_{i_{m-1}} \right\|.$$

For any two indices $m_1 < m_2$ separated by $pN$ indices (where $p > 0$ remains finite), we have

$$\begin{aligned}
u_\ell^T(z_{i_{m_2}} - z_{i_{m_1}}) &\geq \sum_{m=m_1+1}^{m_2} \frac{\left\| W_\ell(z_{i_m} - z_{i_{m-1}}) \right\| + \left\| W_\ell(z_{i_m} - z_{i_{m+1}}) \right\|}{2} \\
&\quad - \sqrt{\frac{c_1 - 2\log|Jf_\theta(x)|_+}{pL}} \sum_{m=m_1+1}^{m_2} \frac{\left\| z_{i_m} - z_{i_{m-1}} \right\| + \left\| z_{i_m} - z_{i_{m+1}} \right\|}{2} \\
&\geq e^{-\frac{\left\| \theta^{(\ell+1:L)} \right\|^2 - (L-\ell)}{2}} \sum_{m=m_1+1}^{m_2} \frac{d_{1,i_m} + d_{2,i_m}}{2} \\
&\quad - (m_2 - m_1) e^{\frac{\left\| \theta^{(1:\ell-1)} \right\|^2 - (\ell-1)}{2}} \sqrt{\frac{c_1 - 2\log|Jf_\theta(x)|_+}{pL}} \text{diam}\Omega \\
&\geq (m_2 - m_1) \left[ c e^{-\frac{\left\| \theta^{(\ell+1:L)} \right\|^2 - (L-\ell)}{2}} N^{-\frac{1}{k^*}} - e^{\frac{\left\| \theta^{(1:\ell-1)} \right\|^2 - (\ell-1)}{2}} \sqrt{\frac{c_1 - 2\log|Jf_\theta(x)|_+}{pL}} \text{diam}\Omega \right],
\end{aligned}$$

where we used the fact that up to a few outliers $\frac{d_{1,i} + d_{2,i}}{2} = \Omega(N^{\frac{1}{k^*}})$.

Thus for $L \geq \frac{4}{c^2} e^{c_1+1} N^{\frac{2}{k^*}} \frac{c_1 - 2\log|Jf_\theta(x)|_+}{p} (\text{diam}\Omega)^2$, we have $u_\ell^T(z_{i_{m_2}} - z_{i_{m_1}}) \geq (m_2 - m_1) \frac{c}{2} e^{-\frac{\left\| \theta^{(\ell+1:L)} \right\|^2 - (L-\ell)}{2}} N^{-\frac{1}{k^*}}$. which implies that at least half of the activations $z_i$ have norm larger than $\frac{c}{8} e^{-\frac{\left\| \theta^{(\ell+1:L)} \right\|^2 - (L-\ell)}{2}} N^{1-\frac{1}{k^*}}$.

This implies that at least one fourth of the indices $i$ satisfy for at least one fourth of the $pL$ layers $\ell$

$$\|\alpha_{\ell-1}(\tilde{x}_i)\| \geq \frac{c}{8} e^{-\frac{\left\| \theta^{(\ell+1:L)} \right\|^2 - (L-\ell)}{2}} N^{1-\frac{1}{k^*}}.$$

Now amongst these indices there are at least some such that there is a point $x$ in the interval $[\tilde{x}_i, \tilde{x}_{i+1}]$ with $\|Jf_\theta(x)\|_{op} \geq cN^{1-\frac{1}{k^*}}$. Since $x$ is $O(N^{-1})$-close to $\tilde{x}_i$ one can guarantee that $\|\alpha_{\ell-1}(x)\| \geq c' e^{-\frac{\left\| \theta^{(\ell+1:L)} \right\|^2 - (L-\ell)}{2}} N^{1-\frac{1}{k^*}}$ for some constant $c'$.

But the Jacobian $J(\tilde{\alpha}_\ell \to f_\theta)$ also explodes at $x$ since

$$\|J(\tilde{\alpha}_\ell \to f_\theta)(x)\|_{op} \geq \frac{\|Jf_\theta(x)\|_{op}}{\|J\tilde{\alpha}_\ell(x)\|_{op}} \geq e^{-\frac{\left\|\theta^{(1:\ell)}\right\|^2 - \ell}{2}} cN^{1-\frac{1}{k^*}}.$$

We can now lower bound the NTK

$$
\begin{aligned}
\mathrm{Tr}\left[\Theta^{(L)}(x,x)\right] &= \sum_{\ell=1}^{L} \|\alpha_{\ell-1}(x)\|^2 \|J(\tilde{\alpha}_\ell \to f_\theta)(x_1)\|_F^2 \\
&\geq \frac{pL}{4} c'^2 e^{-\left\|\theta^{(\ell+1:L)}\right\|^2 + (L-\ell)} N^{2-\frac{2}{k^*}} e^{-\left\|\theta^{(1:\ell)}\right\|^2 + \ell} c^2 N^{2-\frac{2}{k^*}} \\
&= c'' L e^{-c_1} N^{4-\frac{4}{k^*}}.
\end{aligned}
$$

$\square$

This suggests that such points are avoided not only because they have a large $R^{(1)}$ value, but also (if not mostly) because they lie at the bottom of a very narrow valley.

# D Technical Results

## D.1 Regularity Counterexample

We guve here an example of a simple function whose optimal representation geodesic does not converge, due to it being not uniformly Lipschitz:

**Example 20.** The function $f : \Omega \to \mathbb{R}^3$ with $\Omega = [0,1]^3$ defined by

$$f(x,y,z) = \begin{cases} (x,y,z) & \text{if } x \leq y \\ (x,y,z+a(x-y)) & \text{if } x > y \end{cases}$$

satisfies $R^{(0)}(f;\Omega) = 3$ and $R^{(1)}(f;\Omega) = 0$. The optimal representations of $f$ are not uniformly Lipschitz as $L \to \infty$.

*Proof.* While we are not able to identify exactly the optimal representation geodesic for the function $f$, we will first show that $R^{(1)}(f;\Omega) = 0$, and then show that the uniform Lipschitzness of the optimal representations would contradict with Proposition 13.

(1) Since the Jacobian takes two values inside $\mathbb{R}^3_+$, either the identity $I_3$ or $\begin{pmatrix} 1 & 0 & 0 \\ 0 & 1 & 0 \\ 1 & -1 & 1 \end{pmatrix}$, we

know by Theorem 11 that $R^{(1)}(f;\Omega) \geq 2\log|I_3|_+ = 0$. We therefore only need to construct a sequence of parameters of different depths that represent $f$ with a squared parameter norm of order $3L + o(1)$. For simplicity, we only do this construction for even depths (the odd case can be constructed similarly). We define:

$$W_\ell = \begin{pmatrix} e^\epsilon & 0 & 0 \\ 0 & e^\epsilon & 0 \\ 0 & 0 & e^{-2\epsilon} \end{pmatrix} \text{ for } \ell = 1, \ldots, \frac{L}{2} - 1$$

$$W_{\frac{L}{2}} = \begin{pmatrix} 1 & 0 & 0 \\ 0 & 1 & 0 \\ 0 & 0 & 1 \\ \sqrt{a}e^{-\frac{L-2}{2}\epsilon} & -\sqrt{a}e^{-\frac{L-2}{2}\epsilon} & 0 \end{pmatrix}$$

$$W_{\frac{L}{2}+1} = \begin{pmatrix} 1 & 0 & 0 & 0 \\ 0 & 1 & 0 & 0 \\ 0 & 0 & 1 & \sqrt{a}e^{-(L-2)\epsilon} \end{pmatrix}$$

$$W_\ell = \begin{pmatrix} e^{-\epsilon} & 0 & 0 \\ 0 & e^{-\epsilon} & 0 \\ 0 & 0 & e^{2\epsilon} \end{pmatrix} \text{ for } \ell = \frac{L}{2} + 2, \ldots, L$$

We have for all $x \in \mathbb{R}_+^3$

$$\alpha_{\frac{L}{2}-1}(x) = \begin{pmatrix} e^{\frac{L-2}{2}\epsilon}x_1 \\ e^{\frac{L-2}{2}\epsilon}x_2 \\ e^{-(L-2)\epsilon}x_3 \end{pmatrix}$$

and

$$\alpha_{\frac{L}{2}}(x) = \begin{pmatrix} e^{\frac{L-2}{2}\epsilon}x_1 \\ e^{\frac{L-2}{2}\epsilon}x_2 \\ e^{-(L-2)\epsilon}x_3 \\ \sigma(x_1 - x_2) \end{pmatrix}$$

and

$$\alpha_{\frac{L}{2}+1}(x) = \begin{pmatrix} e^{\frac{L-2}{2}\epsilon}x_1 \\ e^{\frac{L-2}{2}\epsilon}x_2 \\ e^{-(L-2)\epsilon}\left(x_3 + \sigma(x_1 - x_2)\right) \end{pmatrix}$$

and

$$f_\theta(x) = \begin{pmatrix} x_1 \\ x_2 \\ x_3 + \sigma(x_1 - x_2) \end{pmatrix}.$$

The norm of the parameters is

$$\frac{L-2}{2}(2e^{2\epsilon} + e^{-4\epsilon}) + (3 + 2e^{-(L-2)\epsilon}) + (3 + e^{-2(L-2)\epsilon}) + \frac{L-2}{2}(2e^{-2\epsilon} + e^{4\epsilon})$$
$$= 3L + 2\left(e^{2\epsilon} - 1\right) + (e^{-4\epsilon} - 1) + 2e^{-(L-2)\epsilon} + e^{-2(L-2)\epsilon} + 2\left(e^{-2\epsilon} - 1\right) + (e^{4\epsilon} - 1)$$

If we take $\epsilon = L^{-\gamma}$ for $\gamma \in (\frac{1}{2}, 1)$, then the terms $2e^{-(L-2)\epsilon}$ and $e^{-2(L-2)\epsilon}$ decay exponentially (at a rate of $e^{L^{1-\gamma}}$), in addition the terms $2\left(e^{2\epsilon} - 1\right) + (e^{-4\epsilon} - 1)$ and $2\left(e^{-2\epsilon} - 1\right) + (e^{4\epsilon} - 1)$ are of order $L^{-2\gamma}$. This proves that $R^{(1)}(f; \Omega) = 0$.

(2) Let us now assume that the optimal representation of $f$ is $C$-uniform Lipschitz for some constant $C$, then by Proposition 13, we have that

$$R^{(1)}(f; \Omega) \geq \log |I_3|_+ + \log \left| \begin{pmatrix} 1 & 0 & 0 \\ 0 & 1 & 0 \\ 1 & -1 & 1 \end{pmatrix} \right|_+ + C^{-2} \left\| I_3 - \begin{pmatrix} 1 & 0 & 0 \\ 0 & 1 & 0 \\ 1 & -1 & 1 \end{pmatrix} \right\|_* > 0,$$

which contradicts with the fact that $R^{(1)}(f; \Omega) = 0$. $\qquad \square$

## D.2 Extension outside FPLFs

Since all functions represented by finite depth and width networks are FPLFs, the representation cost of any such function is infinite. But we can define the representation cost of a function $f$ that is the limit of a sequence of FPLF as the infimum over all sequences $f_i \to f$ converging of $\lim_{i\to\infty} R(f_i; \Omega)$ (for some choice of convergence type that implies convergence of the Jacobians $Jf_i(x) \to Jf(x)$). Note that since the representation cost $R(f; \Omega)$ is lower semi-continuous, i.e. $\liminf_{f \to f_0} R(f; \Omega) \geq R(f_0; \Omega)$, this does not change the definition of the representation cost on the space of FPLFs.

## E Numerical Experiments

For the first numerical experiment, the data pairs $(x, y)$ were generated as follows. First we sample a 8-dimensional 'latent vector' $z$, from which we define $x = g(z_1, \ldots, z_8) \in \mathbb{R}^{20}$ and $y = h(z_1, z_2) \in \mathbb{R}^{20}$ for two random functions $g : \mathbb{R}^8 \to \mathbb{R}^{20}$ and $h : \mathbb{R}^2 \to \mathbb{R}^{20}$ given by two shallow networks with random parameters. Assuming that $g$ is injective (which it is with high probability), the function $f^* = h \circ g^{-1}$ which maps $x$ to $y$ has BN-rank 2.

