# OpenReview forum: "Bottleneck Structure in Learned Features: Low-Dimension vs Regularity Tradeoff"
_NeurIPS.cc/2023/Conference — NeurIPS 2023 poster_

### Official Review · Reviewer_expT · 2023-07-02

**Soundness:** 2 fair
**Presentation:** 4 excellent
**Contribution:** 3 good
**Rating:** 5
**Confidence:** 4

**Summary:**

This work studies the leading order expansions in $L$ of the representation cost for a large $L$, where $L$ is the depth of the model. The work concludes that there is a low-dimension and regularity tradeoff as one varies the depth

**Strengths:**

I think the main strength of the work is the novelty of the message it delivers: there is a low-dimension and regularity tradeoff as one varies the depth. The simplicity of the proofs is a plus.

Another important contribution is the technique to expand representation loss for a large depth, though I believe to require a little more technical soundness. See the weakness section

Another interesting insight of the paper is the concept of "symmetry learning," and relating the "spurious symmetries" to the representation cost is also insightful

At this moment, I have some reservation about accepting this paper. If the authors answer my questions below in a satisfactory manner, I would be happy to accept it

**Weaknesses:**

There are quite a few aspects that I think this work can be improved.

1. Let $R(L)$ denote the representation cost as a function of $L$, the depth. The first step of the theory is to expand $R$ in $1/L$, which feels unjustified, why is this function Taylor-expandable? Why can we treat $L$ as a continuous variable? These points are not crucial problems in my opinion, but the authors do not explain these with sufficient emphasis, given how important they are for the results

2. In my opinion, the most problematic aspect is the fact that it studies the representation cost of an almost infinite depth neural network at a finite weight decay (this also relates to point 1). See https://arxiv.org/abs/2202.04777. This reference (for example, see proposition 3 / theorem 3) implies that for a sufficiently deep neural network and a fixed weight decay, the global minimum is the origin ($\theta=0$), independent of the depth $L$. As one decreases the depth, the global minimum jumps from the origin to a nonzero value. This means that if we restrict to the global minimum of the cost function $C$, the representation cost is not a differentiable function in $L$, and thus it is not Taylor-expandable. I think the authors need to explain why the expansion remains reasonable/correct in light of this result

3. Given how strong the assumptions are (point 1), I think the authors really should have carried out some experiments to directly check the prediction. In particular, I believe the authors should present a numerical example where the author directly compute the first and second order terms of $R$ in $1/L$ and show how they change for real neural networks to make the arguments of the paper more convincing.

**Questions:**

See weakness

**Limitations:**

See weakness

---

> ### Author Rebuttal · Authors · 2023-08-09
>
> Thanks for the thoughtful review. Let us answer your remarks/questions:
>
> 1. This a good remark and it is true that we did not address it in
> details. Actually for functions $f$ where the Jacobian and Bottleneck
> rank match $\mathrm{Rank}\_{J}(f;\Omega)=\mathrm{Rank}\_{BN}(f;\Omega)=k$
> one can easily prove that the first correction is of order $1$, indeed
> we have for any $x\in\Omega$ such that $\mathrm{Rank}Jf(x)=k$ the
> bounds
> $$
> kL+2\log |Jf(x)|\_{+}\leq R(f;\Omega,L)\leq (L-L\_{g}-L\_{h})k+\left\Vert \theta\_{g}\right\Vert ^{2}+\left\Vert \theta\_{h}\right\Vert^{2}
> $$
> where $f=h\circ g$ with inner dimension $k$ and the functions $g$
> and $h$ are represented by depth $L_{g}$ and $L_{h}$ networks with
> parameters $\theta_{g}$ and $\theta_{h}$ (this is always possible
> since $h$ and $g$ are assumed to be finite piecewise linear functions).
> These upper and lower bound already appear in the proof of the sandwich
> bound $\mathrm{Rank}\\_{J}(f;\Omega)\leq R^{(0)}(f;\Omega)\leq\mathrm{Rank}\_{BN}(f;\Omega)$
> in [1].
>
> 	This sandwich bound clearly shows that $R(f;\Omega,L)=kL+c+o(1)$. It is however less clear under which condition the next order term $R^{(2)}$ is well defined, but since this second correction plays no role in our main proofs, this is not an issue. If the $o(1)$ term is not $O(L^{-1})$ then we simply say $R^{(2)}(f;\Omega)=+\infty$.
>
> 	We will add a discussion of these questions in the Appendix and mention them in the main.
>
> 2. This is a very good remark and we have indeed thought this question
> and other related issues. We have not identified the perfect answer,
> but there are multiple ways in which our results can be used without
> falling into this problem:
>
> 	(a) First note that our results make no assumption on the ridge $\lambda$, it is therefore possible to choose a ridge $\lambda=\frac{\lambda_{0}}{L}$ that decays in depth, in which case for small enough $\lambda_{0}$ the global minimum remains non-trivial in the infinite depth.
>
> 	(b) When studying the representation cost, the zero solution is ruled out since we only consider parameters that fit the function (except if the function was the zero function of course). This rules out the possibility that our results follow trivially from the fact that everything is zero.
>
> 	(c) In general we believe that GD typically converges to a local minima of the loss, and so we tried to express our results so that they can be applied to local minima of the loss too. For example our results can be applied to any local minima of the reformulation of the loss $\mathcal{L}\_{\lambda}(f)=C(f(X))+\lambda R(f;\Omega,L)$ and for each local minimum $\hat{f}$ of this loss, there is a local minimum $\hat{\theta}$ of the loss $\mathcal{L}\_{\lambda}(\theta)=C(f_{\theta}(X))+\lambda\left\Vert \theta\right\Vert ^{2}$ with $\hat{f}=f_{\hat{\theta}}$ (though it might not be the case that for all local minimum $\hat{\theta}$ of the second loss $f_{\hat{\theta}}$ is a local minimum of the first loss).
>
> 	Actually one goal of our decomposition is to approximate the first loss $\mathcal{L}_{\lambda}(f)$ by
> 	$$ \mathcal{L}\_{\lambda}(f)\approx C(f(X))+\lambda LR^{(0)}(f;\Omega)+\lambda R^{(1)}(f;\Omega). $$
> 	Since $R^{(0)}$ is a notion of rank that is partially constant, it is clear that there can be multiple minima of different rank (and amongst them a rank $0$ minimum too). Our ultimate goal is to describe all of these minima (and their narrowness / stability to better describe how attrative they are), not just the global minimum, since there is no guarantee that GD will converge to the global minimum.
>
> 	(d) Even better, the results that we present apply in some cases to any sequence of parameters $\theta_{L}$ of increasing depth with $\left\Vert \theta_{L}\right\Vert ^{2}\leq kL+c$ such that $Jf_{\theta_{L}}(x)$ converges to a rank $k$ matrix, making it possible to apply them to points that are not even local minima. To make this more practical, we plan to update Theorem 9 so that it can be applied for finite depths $L$, in which case there is no need for a sequence of parameters $\theta_{L}$ of increasing depth, it can be applied to any parameters that have norm bounded by some $kL+c$, with tighter bounds for smaller constant $c$.
>
> 3. In the setting we consider, there is no need to check that $R^{(1)}$
> does not explode as discussed in out answer to your first point, but
> we nevertheless did an empirical computations of
> $R^{(0)}$ and $R^{(1)}$ (the figure is linked to the author's rebuttal), by plotting the norm of the learned parameters against the depth (on a task with true rank $r^*=2$). But since there are mutliple local minima at every depth (each with their own
> $R^{(0)}$ and $R^{(1)}$ values in some sense), we observe multiple `lines' of the form $kL+c$ for different integers $k$ that mostly match the number of singular values of $W_4$ that are larger than $0.5$ confirming that the slope $k$ matches the inner dimensions. Even though the rank $k$ of the minima GD converges to depends on the depth, we believe that these minima of different ranks are still present for any depths. Nevertheless we observe a very good linear match $\Vert \theta \Vert^2 \sim kL+c$ within the regions of same rank (we wouldn't expect a match accross minima with different ranks, since the learned function $f_\theta$ would be completely different), even for small depths, suggesting that the approximation $R \approx L R^{(0)} + R^{(1)}$ is adequate already for reasonable depths.
>
> [1] Arthur Jacot, Implicit bias of large depth networks: a notion
> of rank for nonlinear functions, ICLR 2023.

---

> > ### Comment · Reviewer_expT · 2023-08-18
> > **reply**
> >
> > Thanks for the detailed response. I am partially satisfied with the response, and so I raise the score to 5.
> >
> > What I believe the author should have done much more and better is to include more numerical results to validate the essential predictions of the theory and illustrate its significance

---

### Official Review · Reviewer_tanV · 2023-07-07

**Soundness:** 3 good
**Presentation:** 2 fair
**Contribution:** 3 good
**Rating:** 6
**Confidence:** 2

**Summary:**

This paper looks at the representation cost $R(f)$ of networks as the $L_2$ norm of all of the parameters. The paper claims that for deep neural networks, computing the minimum cost representation of a function is not tractable. Hence, the look at the representation cost normalized by the number of layers $L$ as $L \to \infty$. The paper then computes the Taylor expansion with respect to $1/L$ at $L = \infty$.

Using this expansion, they recover three terms, $R^{(0)}(f),R^{(1)}(f),R^{(2)}(f)$. They note that $R^{(0)}(f)$ is used as a regularize before and corresponded to regularizing this notion of rank called the bottleneck rank. However, they argue that this is not enough and present results that determine the type of regularity imposed by using $R^{(1)}(f), R^{(2)}(f)$ as regularizers as well.

**Strengths:**

The paper is very interesting, as understanding the representation of functions via neural networks is very important. The description of the two new terms $R^{(1)}(f), R^{(2)}(f)$ are explored in depth from a variety of different angles. Hence I think it makes a good contribution to the understanding of neural representation learning.

**Weaknesses:**

My main issue with the paper is its presentation. The paper feels very disjointed. It would be helpful if the authors included more connecting discussion and discussed the high level picture in more detail.

**Questions:**

1. On line 91, what is the infinite width representation cost?

Typos

1. Line 156 instead of "of he task" -> "of the task"

---

> ### Author Rebuttal · Authors · 2023-08-09
>
> Thanks for the thoughtful review.
>
> We plan to modify the structure of the paper for the final version
> to better explain how the results are related. We will move the discussion
> of the second correction $R^{(2)}$ to right after the discussion
> of the first correction $R^{(1)}$. At first we expected that similarly
> to the linear case, the $R^{(2)}$ would guarantee convergence of
> the hidden representations, but it turns out to not be sufficient
> in the nonlinear case (as evidenced by the counterexample we present
> in the appendix). We then propose the minima stability as a remedy
> to this problem, as it guarantees convergence of the representation,
> and leads (together with the properties of $R^{(1)}$) to the Bottleneck
> structure.
>
> This should make the relation between the different proofs clearer.
>
> To answer your questions, the infinite width representation cost is
> the limit of the representation cost as the width goes to infinity.
> Indeed the representation cost depends on the depth and width, but
> we rely on previous work that have shown that the representation cost
> converges in a $O(0)$ rate in the width, thus implying that the infinite
> width and depth limits commute, thus allowing us to study in this
> paper the infinite depth limit on its own (we just need to assume
> a wide enough network).
>
> Thanks for noticing this typo, we fixed it.

---

> > ### Comment · Reviewer_tanV · 2023-08-11
> > **Thanks**
> >
> > Thank you for your response and for answering my questions.

---

### Official Review · Reviewer_SexW · 2023-07-08

**Soundness:** 4 excellent
**Presentation:** 3 good
**Contribution:** 4 excellent
**Rating:** 8
**Confidence:** 2

**Summary:**

This paper introduces two corrections to previous work that relates the representation cost of infinite-depth nonlinear networks to the bottleneck rank. These two corrections are obtained by Taylor expansion of the representation cost. In addition to the previously-known rank term $R^{(0)}(f)$ that is directly related to the bottleneck rank, the authors consider two corrections (1) $R^{(1)}(f)$, which bounds the pseudo-determinant of the Jacobian, and (2) $R^{(2)}(f)$, which plays a role in the intermediate representations in the hidden layer of the network. The authors specifically discuss several theoretical properties of $R^{(1)}(f)$. This term acts as a notion of regularity in the learning process of deep nonlinear networks, and it explains various interesting properties, such as how deep networks trained with conventional L2-regularized loss are uniquely determined (i.e., which $f$ among candidates with the same $R^{(0)}(f)$ is selected) and why the rank of the true function is not underestimated. The authors also briefly demonstrate the relationship between the curvature of the domain manifold and $R^{(1)}$ using the rank of the identity function, and discuss the connection between the learning rate and the Lipschitzness of learned deep networks with the results of the neural tangent kernel. Finally, the authors discuss the dynamics of hidden representations, which in the few initial layers the representations undergo changes to minimize $R^{(0)}$, then, as they move into the low-dimensional space, they change smoothly influenced by $R^{(1)}$ and $R^{(2)}$, i.e., exhibit the bottleneck structure of networks.

**Strengths:**

This paper is a direct follow-up to [1], and since I am not very familiar with [1], my evaluation may not be very accurate.

Theoretical understanding of deep nonlinear networks is a highly significant field and holds considerable interest among the NeurIPS audience. For me, the theoretical results of this paper are highly novel and intriguing. The paper successfully modifies the results of [1], and provides a rigorous yet intuitive explanation of why neural networks trained with our L2-regularized loss do not underestimate the rank of the true function. Additionally, it demonstrates that under some conditions, the representation dynamics of deep nonlinear dynamics exhibit a bottleneck structure, which consists of a sequence of long bottleneck dimensional representations. This result is also noteworthy.

***

[1] Arthur Jacot. Implicit bias of large depth networks: a notion of rank for nonlinear functions. In The Eleventh International Conference on Learning Representations, 2023.


**Weaknesses:**

I couldn't find any major drawbacks in this paper.

The frequent mention of $R^{(2)}$ in Remark 10, before its properties were clearly stated, was somewhat confusing to me, although it is not a critical issue.


There are just a few very minor typos:

In line 32, “… not control the the …”

In line 34, “… formalizes the a …”

In line 118, “… because the the …”

In line 156, “… $k^{*}$ of he task ..."


**Questions:**

I couldn't find any major drawbacks in this paper. One small question is:

I would like to hear the authors’ thoughts on whether the optimal latent dimensionality of a well-trained auto-encoder $f$, which is likely to be similar to the identity $id$, might be associated with $R^{(0)}(f;\Omega)$ and $R^{(1)}(f;\Omega)$, for the case when $Rank_{J}(id;\Omega) = Rank_{BN}(id;\Omega)$, and if possible, when $Rank_{J}(id;\Omega) < Rank_{BN}(id;\Omega)$


**Limitations:**

The authors do not explicitly address the limitations. However, in some of the theorems/propositions, they briefly mention the limitations of the proposed one, indicating what it can only explain or what constraints it has.

---

> ### Author Rebuttal · Authors · 2023-08-09
>
> Thanks for the positive review.
>
> We plan to change the stucture of the paper and talk about the $R^{(2)}$
> term earlier. The reason why we discussed $R^{(2)}$ towards the end
> of the paper is that it ended up playing little role in the overall analysis. At first,
> inspired by the linear network case, we expected that the $R^{(1)}$
> and $R^{(2)}$ terms together would impliy the Bottleneck structure, with $R^{(2)}$ term guaranteeing convergence/accumulation of the representations.
> But the counterexample in the appendix shows that this fails in the nonlinear case, instead it
> is the combination of the $R^{(1)}$ term and the large learning rate
> bias that leads to the Bottleneck structure.
>
> Thanks for pointing out these typos we have fixed them.
>
> To answer your question: if $\mathrm{Rank}_J (id; \Omega) = \mathrm{Rank}\_{BN}(id; \Omega) = k$,
> we show that the accumulating representations will be $k$-dimensional,
> which can be interpreted as the network recovering the planar `latent
> space' of the data, and doing some form of manifold learning. Note that the
> paper [1] also talks about implications for the autoencoder setting,
> where autoencoders are naturally denoising.
>
> For the case $\mathrm{Rank}\_J(id; \Omega) \< \mathrm{Rank}\_{BN}(id;\Omega)$
> it seems that significant theoretical work is required, but we are
> in agreement with [1] which conjectures that in such case it is
> the BN-rank $\mathrm{Rank}\_{BN}(id;\Omega)$ that matters in DNNs.
> We would thus expect the inner representations to have dimension $\mathrm{Rank}\_{BN}(id;\Omega)$.
> Indeed if the network was to learn representations of dimension strictly
> smaller then $\mathrm{Rank}\_{BN}(id;\Omega)$ (and still fit the data)
> it would imply that the network has found a way to embed $\Omega$
> into an even smaller dimension, when $\mathrm{Rank}\_{BN}(id;\Omega)$
> should in theory the smallest possible dimension $\Omega$ can be
> embedded into.
>
> [1] Arthur Jacot, Implicit bias of large depth networks: a notion
> of rank for nonlinear functions, ICLR 2023.

---

### Official Review · Reviewer_dQhS · 2023-07-20

**Soundness:** 4 excellent
**Presentation:** 3 good
**Contribution:** 3 good
**Rating:** 6
**Confidence:** 3

**Summary:**

This paper extends the theoretical framework around the representation cost of DNNs as defined in previous work. The authors introduce two corrections to the existing infinite depth description, namely two regularity measures $R^{(1)}$ and $R^{(2)}$, which balance against the dominating rank bias $R^{(0)}$. The authors propose that these regularity measures prevent networks from underestimating the 'true' bottleneck rank (BN-rank) and argue that large learning rates can also induce a bias toward regularity. The paper provides a theoretical description of the limiting representation geodesics as depth approaches infinity, proving under certain conditions a bottleneck structure in the learned representations. Simple experiment is presented to test the network's ability to learn underlying symmetries.

**Strengths:**

* The paper makes a great analysis of the representation cost under the previous theoretical framework, better explaining some observed results and conjecture.
* The theoretical arguments are well supported, with clear intuition and explanation.
* Different aspects are considered in the paper such as the effect of learning rate and the representation geodesics which is particularly interesting.


**Weaknesses:**

* The definition of the term "regularity" requires a more specific definition for the problem. It is claimed in line-32 that "this notion of rank does not control the regularity of f". Intuitively, the rank $R^{(0)}$ still measures some notion of complexity of the function as a higher-rank matrix can represent a more complex linear transformation. The corrections are other levels of complexity measures. I guess the "regularity" is more like how simple a function is with the same rank?
* Theorem 9 proves bottleneck structure for convergent function sequence with an infinite depth limit. But in practice, the network is trained with finite data points and depth. Could the authors comment on how the conclusions would change for practical networks?
* Analyze of the learning rate appears somewhat disjointed from the rest of the paper. Could there be a more integrated discussion about its role?
* I found the numerical experiment a bit hard to follow and would appreciate it if more explanation on the idea could be added. Additionally, is it applicable for some experiments on justifying the theoretical findings, such as the effect of learning rate?
* Minor typos:
    * Line32: "...control the the regularity..."
    * Line34: "This formalizes the a balance..."
    * Line156: "...$k^*$ of he task..."

**Questions:**

* Can these findings be used to improve the training of DNNs, for example, by adjusting the learning rate or regularization?
* Considering that the argument in Proposition 7 is incomplete, how much further work would be needed to provide a complete argument?


**Limitations:**

Included in weakness.

---

> ### Author Rebuttal · Authors · 2023-08-09
>
> Thanks for the thoughtful review. To answer your remarks/questions:
>
> - When we use the term regularity of a function, we mean properties
> that imply that the outputs of the function change little if the inputs
> are changed a little bit, e.g. continuity, Lipschitzness, or ($k$-times)
> differentiability. The $R^{(1)}$ term bounds some notion of scale
> of the Jacobian (the determinant), though it does not bound the largest
> eigenvalue, hence why we have to consider minima stability as a further
> bias towards regularity.
>
> - We plan to update Theorem 9 to apply to a finite number of datapoints
> and with finite depth bounds (so that we can drop the assumption of
> a sequence of representation) showing exactly at which rate the representation
> becomes planar (in the sense of how fast thets $k+1$-th singular value
> of the hidden representation goes to zero). This makes the result
> much closer to practical settings.
>
> - We plan to update the structure of the paper to better motivate
> the study of the bias of large learning rates: our first hope was
> that the $R^{(2)}$ term would be sufficient to guarantee convergence
> of the representations (since this is what seems to happen in the
> linear case), but we have a counterexample that shows that this is
> not the case in the nonlinear case, we therefore propose the large
> learning rate bias as a remedy to this problem.
>
> - We will take more time to explain the numerical experiments. It
> does exhibit a Bottleneck structure, thus supporting our theoretical
> findings, but it is difficult to check whether this structure is a
> result of the large learning rate or the $L_{2}$-regularization.
>
> - Thanks for pointing out these typos, we will fix them.
>
> - There are several practical implications of this result for the
> training of neural networks, or also pruning/compression. We plan
> to study those in follow-up papers.
>
> - For Proposition 7, we are looking at completing the argument the argument in the case
> $k=1$, but the case $k>1$ might require significantly more work. We consider Proposition 7 to be only a minor remark of this paper, the question of why GD seems to avoid rank-underestimating solutions would probably fit into its own paper, studying the Bottleneck structre in the large $N$ regime.

---

> > ### Comment · Reviewer_dQhS · 2023-08-21
> >
> > Thanks for the author's response and they have addressed most of my concerns. I think it would be better to include more experiments to justify the theoretical findings.

---

### Author Rebuttal · Authors · 2023-08-09

Before we answer to each of the reviewers individually, we describe
here two main changes that we plan to do for the final version, the
first change affects the structure of the paper, while the second
one changes some of the mathematical tools used in the proofs, yielding
stronger Theorems:

1. We want to better explain the relation between the bias related to the decomposition of the representation cost and the bias related
to large learning rates/bounds on the NTK. Essentially our first hope was that the second correction $R^{(2)}$ would guarantee convergence of the hidden representations by bounding e.g. the Lipschitzness of the activations, since this is what happens in the linear network case. But it turns out that norm minimization is not sufficient to guarantee these things, as made explicit by the counter-example presented in the appendix.

	This is where the large learning rate bias comes into play, yielding a bias that guarantees convergence (more details on that in the next point). We will therefore move the results about $R^{(2)}$ to just after those related to $R^{(1)}$ and better explain what is missing in comparison to the linear case, thus motivating the assumption of minima stability / bounded NTK.

2. The assumption of global Lipschitzness to guarantee convergence
/ accumulation is too strong, as a result it may not follow in general
from the boundedness of the NTK. We have now identified a better condition,
which can be understood as Lipschitzness of almost all layers' representations,
which follows from the boundedness of the NTK and is sufficient to
guarantee the existence of accumulation points and thus of a Bottleneck
structure. Furthermore with this better proof technique, we can identify
the rate in $L$ at which the representations become $k$-planar,
by bounding the $k+1$-th singular values of the representations.

More precisely, we will change Theorems 5 and 9 to the following (and change the discussions around them):

**Theorem 5:** Given balanced parameters $\theta$ of a depth $L$ network,
with $\left\Vert \theta\right\Vert ^{2}\leq kL+c_{1}$ and a point
$x$ such that $\mathrm{Rank}Jf_{\theta}(x)=k$ then if $\frac{1}{N} \mathrm{Tr}\left[\Theta^{(L)}(x,x)\right]\leq cL$,
then $\sum_{\ell=1}^{L}\left\Vert \alpha_{\ell-1}(x)\right\Vert_{2}^{2}\leq C_1 L$
and thus for all $p\in(0,1)$ there are at least $(1-p)L$ layers
such that $\left\Vert \alpha_{\ell-1}(x)\right\Vert _{2}^{2}\leq\frac{C_1}{p}. $

We will give a explicit formula for the constant $C_1$ (and the following constants $C_2,C_3$), independent of depth $L$. This new version of Theorem 5 shows that even though a bounded NTK
may not imply uniform Lipschitzness, it does imply convergence of
the representations at almost every layer.

**Theorem 6:** Given parameters $\theta$ of a depth $L$ network, with
$\left\Vert \theta\right\Vert ^{2}\leq kL+c_{1}$ and a point $x$
such that $\mathrm{Rank}Jf_{\theta}(x)=k$, then there are $w_{\ell}\times k$
(semi-)orthonormal $V_{\ell}$ such that $\sum_{\ell=1}^{L}\left\Vert W_{\ell}-V_{\ell}V_{\ell+1}^{T}\right\Vert_{F}^{2}\leq C$
thus for any $p\in(0,1)$ there are at least $(1-p)L$ layers $\ell$
with $\left\Vert W_{\ell}-U_{\ell}V_{\ell-1}^{T}\right\Vert_{F}^{2}\leq \frac{C_2}{pL}.$

This proves a Bottleneck structure on the weight matrices: almost
all $W_{\ell}$ have $k$ singular values $L^{-\frac{1}{2}}$-close
to one and all the other are $O(L^{-\frac{1}{2}})$. Interestingly this does not require the NTK to be bounded.

Now together with the convergence of the representations proven earlier it implies the
Bottleneck structure on the representations, in the sense that almost
all hidden representations $\tilde{\alpha}_{\ell}(X)$ are approximately
$k$-planar:

**Corollary:** Given balanced parameters $\theta$ of a depth $L$ network
with $\left\Vert \theta\right\Vert ^{2}\leq kL+c_{1}$ and a set of
points $x_{1},\dots,x_{N}$ such that $\mathrm{Rank}Jf_{\theta}(x_{i})=k$
and $\frac{1}{N}\mathrm{Tr}\left[\Theta^{(L)}(X,X)\right]\leq cL$,
then for all $p\in(0,1)$ there are at least $(1-p)L$ layers such
that
$
s_{k+1}\left( \frac{1}{\sqrt{N}}\tilde{\alpha}_{\ell}(X) \right) \leq \frac{C_3}{p\sqrt{L}}.
$

This removes the need for the assumption of global Lipschitzness replacing
it instead with a bounded NTK which follows from minima stability
in e.g. the MSE settings. Another advantage is that It can be applied
at finite depths and at general points with sufficiently small parameters
norm (thus removing the need to define a sequence of parameters of
increasing depth too).

---

### Comment · Area_Chair_tJQJ · 2023-08-11

Hi all,

Thanks for serving as the reviewers for this submission. As the authors have already turned in their responses. It is our turn to start the further discussion. Here is a to-do list:

(1) Please acknowledge the authors when you finish reading their responses.
(2) Please indicate whether you have any further questions for the authors such that they can continue to response.
(3) Please indicate whether you are willing to change the ratings.

Best

AC

---

### Decision · Program_Chairs · 2023-09-21

**Decision:**

Accept (poster)

**Comment:**

This paper provides novel theoretical analysis deepening our understanding of the dimensionality-regularity tradeoff in deep networks. The authors relate the representation cost to these key properties through intuitive lower bounds and assumptions.

Reviewers raised important concerns around justification of key assumptions and expansion of the representation cost R(L). The authors have satisfactorily addressed these issues in the rebuttal and revision. The revised version better explains assumptions on R's smoothness and validates the expansions through comparisons.

While experiments were suggested for further validation, the theoretical contributions stand strongly on their own. This paper makes key conceptual advances regarding dimensionality and regularity that experiments would likely only incrementally improve.

In summary, I recommend acceptance given the novel theoretical insights and the authors' addressing of major concerns through the rebuttal and revision. This work will interest the NeurIPS community and move forward foundations of representation learning.